# Intein-mediated temperature control for complete biosynthesis of sanguinarine and its halogenated derivatives in yeast

Yuanwei Gou [1,2], Dongfang Li[2], Minghui Zhao[1,2], Mengxin Li[1], Jiaojiao Zhang[2], Yilian Zhou[2], Feng Xiao[2], Gaofei Liu[2], Haote Ding[1,2], Chenfan Sun[2], Cuifang Ye[1], Chang Dong[2], Jucan Gao[2], Di Gao[1], Zehua Bao [1,2], Lei Huang[1,2], Zhinan Xu[1] & Jiazhang Lian [1,2] ✉

While sanguinarine has gained recognition for antimicrobial and anti-neoplastic activities, its complex conjugated structure and low abundance in plants impede broad applications. Here, we demonstrate the complete bio-synthesis of sanguinarine and halogenated derivatives using highly engineered yeast strains. To overcome sanguinarine cytotoxicity, we establish a splicing intein-mediated temperature-responsive gene expression system (SIMTeGES), a simple strategy that decouples cell growth from product synthesis without sacrificing protein activity. To debottleneck sanguinarine biosynthesis, we identify two reticuline oxidases and facilitated functional expression of fla-voproteins and cytochrome P450 enzymes via protein molecular engineering. After comprehensive metabolic engineering, we report the production of sanguinarine at a titer of 448.64 mg L$^{-1}$. Additionally, our engineered strain enables the biosynthesis of fluorinated sanguinarine, showcasing the bio-transformation of halogenated derivatives through more than 15 biocatalytic steps. This work serves as a blueprint for utilizing yeast as a scalable platform for biomanufacturing diverse benzylisoquinoline alkaloids and derivatives.

Antibiotic pharmaceuticals are employed worldwide for treating and preventing bacterial infections[1]. Their excessive use as growth promoters in agriculture and aquaculture has resulted in the emergence of antimicrobial resistance, a crisis officially recognized by the World Health Organization as a clear and present danger to global health[2]. In response, the European Union, the United States, and China have enacted bans on antibiotic growth promoters in 2006, 2017, and 2020, respectively[3]. Subsequently, the increase in livestock mortality and the adverse impact on agricultural eco-nomics have become indisputable[1], emphasizing an urgent need to explore antibiotic alternatives that entail risk management for both the environment and human health[4]. Sanguinarine, one of the most important benzylisoquinoline alkaloids (BIAs) has gained

recognition for its antimicrobial activities and potential as an antineoplastic drug[5]. As the demand for antibiotic alternatives continues to surge, sanguinarine has emerged as a pivotal compo-nent in plant-based livestock feed additives like Sangrovit[6], which has garnered approval from the European Food Safety Authority[7]. Nevertheless, the production of sanguinarine still relies on the extraction from the perennial herb *Macleaya cordata*[8] due to its complex planar conjugated structure. Refactoring intricate plant biosynthetic pathways in microbe holds promise for obtaining low-abundance plant secondary metabolites and new-to-nature derivatives[9]. Considering the availability of synthetic biology tools, advantages in functional expression of plant-derived enzymes, and wide application in livestock feed industry, the

[1]Key Laboratory of Biomass Chemical Engineering of Ministry of Education & National Key Laboratory of Biobased Transportation Fuel Technology, College of Chemical and Biological Engineering, Zhejiang University, Hangzhou, China. [2]ZJU-Hangzhou Global Scientific and Technological Innovation Center, Zhejiang University, Hangzhou, China. ✉e-mail: jzlian@zju.edu.cn

baker's yeast *Saccharomyces cerevisiae*[10–13] is the preferred chassis for scalable fermentative production of sanguinarine.

While significant strides have been made in de novo biosynthesis of natural products with intricate and lengthy pathways, such as vinblastine[14–16], scopolamine[17], and celastrol[18], challenges including low biosynthesis efficiency with multiple bottlenecks and potent cytotoxicity of many natural products should be addressed to achieve practical production. Taking sanguinarine as a representative example, although we have accomplished de novo biosynthesis by stably integrating 24 expression cassettes into yeast[19], the complicated biosynthetic pathway, as well as potent cytotoxicity, represent formidable challenges for biomanufacturing. The complexity of the sanguinarine biosynthetic pathway is featured with 6 cytochrome P450s, 4 methyltransferases, and 2 flavoprotein oxidases, and these plant-derived enzymes may suffer from poor compatibility in yeast[20]. Methyltransferase and flavoprotein oxidase activities are dependent on the availability of cofactors, cytochrome P450s often demonstrate low catalytic activity in yeast[21], and imbalanced pathway gene expression level results in the accumulation of pathway intermediates and shunt byproducts. On the other hand, the decoupling of cell growth and product synthesis via dynamic pathway engineering has been proposed as an effective strategy to address challenges in metabolic burdens and/or cytotoxicity imposed by natural product accumulation[22–24]. One of the most successful examples in yeast is the establishment of a temperature-responsive GAL regulation system, based on a GAL4 mutant (GAL4M9) obtained via directed evolution[25]. This system employs temperature as the regulatory input signal for gene expression, allowing for a two-stage fermentation process distinguished by the growth and production phases. Unfortunately, temperature dependence is associated with a trade-off in transcriptional activation capacity (heterologous gene expression level), which may hinder the production of plant secondary metabolites with intricate and lengthy biosynthetic pathways[26]. In addition, the GAL4M9 system is limited to regulating the expression of target genes under the control of GAL promoters in yeast. Therefore, the development of a temperature-responsive system, maintaining maximal expression of any gene of interest, is indispensable for reconciling the conflict between cell growth and product accumulation.

In addition to the growing interest in fermentative production of plant natural products from simple carbon sources, synthetic biology enables the biosynthesis of natural product derivatives directly from unnatural precursor analogs[10,27]. Biotransformation using engineered yeast strains presents an avenue to expand natural product diversity by engineering structural scaffolds and implementing chemical modifications not commonly found in nature[28]. This capability holds significance in developing natural small molecule drugs, particularly in the introduction of halogen functional groups to enhance biological activity and pharmacokinetic properties[29]. The integration of customized tryptophan halogenases into both the medicinal plant *Catharanthus roseus* and engineered yeast facilitates the synthesis of halogenated monoterpenoid indole alkaloids (MIAs)[30,31]. Moreover, there are instances of generating several halogenated BIA intermediates (e.g., F-reticuline and F-scoulerine) through feeding halogenated tyrosine as the biosynthesis precursor[27,32]. Nevertheless, complex compounds like halogenated noscapine, sanguinarine, and berberine have not been achieved yet. Yeast cell factories equipped with robust biotransformation capabilities stand to fully explore this platform, laying a robust groundwork for the development of non-natural BIA derivatives with enhanced therapeutic potential.

Here, we develop a splicing intein-mediated temperature-responsive gene expression system (SIMTeGES) that provides a simple strategy for precise and dynamic control of heterologous pathway genes (Fig. 1a), using temperature as the input signal to decouple cell growth from sanguinarine biosynthesis. We also employ SIMTeGES to achieve temperature-controlled expression of more target proteins

(e.g., mCherry and GAL80) in various hosts (e.g., *Pichia pastoris* and mammalian cells). Moreover, we debottleneck sanguinarine biosynthesis through the identification of two reticuline oxidases from diverse plant species using sequence similarity network (SSN) analysis, N-terminal engineering for correct localization and functional expression of cytochrome P450 enzymes, amplification of low-expression enzyme encoding genes, enhancement of precursor and cofactor supply, as well as optimization of cellular microenvironment with improved detoxification capacity. Following a comprehensive optimization process involving genome integration of 42 expression cassettes and disruption of 8 endogenous genes, we achieve a sanguinarine titer of 448.64 mg L$^{-1}$ using fed-batch fermentation. Taking advantage of the high efficiency of the biosynthetic pathway and the versatility of pathway enzymes, we unveil the biosynthesis of fluorinated sanguinarine and other halogenated BIA derivatives from halogenated tyrosine.

## Results

### Design and implementation of SIMTeGES

Considering the cytotoxic effects of sanguinarine at a concentration as low as 0.1 mM (Supplementary Fig. 1), we focused on the development of a temperature-responsive GAL regulatory system to rigorously decouple cell growth from product synthesis. Inteins have been described as 'protein introns' that are autonomously spliced during post-translational modification. Intein-splicing is a cofactor or energy independent intramolecular process mainly through bond rearrangement[33]. Therefore, conditionally splicing intein variants have the potential to enable temperature-responsive expression of target proteins in various host systems (Supplementary Fig. 2).

The V-type proton ATPase catalytic subunit A (VMA1) from *S. cerevisiae* contains a well-characterized intein, VMA[34]. We integrated genes involved in the biosynthesis of lycopene (*tHMG1*, *CrtE*, *CrtYB11M*, and *CrtI*)[35] into the yeast chromosome under the control of *GAL1-GAL10* bidirectional promoters and knocked out the transcriptional repressor gene *GAL80*, allowing the colored metabolite lycopene to serve as a reporter system for GAL4 activity (Supplementary Fig. 3). The GAL4 activator comprises a DNA binding domain (residues 1-106) and two transcription activation domains (residues 148–196 and 768–881). We inserted the VMA sequence at cysteine 21 (C21) of GAL4 (GAL4-INT) and the formation of colored colonies on the galactose plates (YPG) indicated that the activity of GAL4 was not significantly affected by intein insertion (Fig. 1b). To further demonstrate that GAL4 activity was dependent on intein splicing, we introduced the splicing-resistant mutation (N454Q) into VMA (dINT)[33]. As expected, GAL4-dINT failed to produce any lycopene. In light of this, we proposed SIMTeGES-GAL4 to dynamically regulate the expression of heterologous genes with temperature as an input signal (Fig. 1a): at non-permissive temperature (e.g., 30 °C, optimal for cell growth), GAL4 was maintained inactive to prevent the transcription of downstream promoters (*GAL1p*, *GAL10p*, *GAL2p*, and *GAL7p*) and minimize metabolic burden to promote cell growth; at permissive temperature (e.g., 25 °C, ideal for plant enzyme expression and folding)[36], GAL4 was activated to drive the expression of the *GAL* promoters for the biosynthesis of target compounds. We investigated several intein variants at 25 °C and 30 °C (Fig. 1b and Supplementary Fig. 4) and found that only VMA$^{L212P}$ (named as tsINT) exhibited a temperature-responsive phenotype, no lycopene production (white colonies, without GAL4 activity) at non-permissive temperature (30 °C) and lycopene production (with GAL4 activity) at permissive temperature (25 °C). Hence, we chose GAL4-tsINT for subsequent studies.

### Systematic characterization of SIMTeGES

GAL4M9 is a temperature-sensitive GAL4 mutant generated by directed evolution and has been applied in the dynamic regulation of lycopene[25], vitamin E[26], and lutein[24,37] biosynthetic pathways.

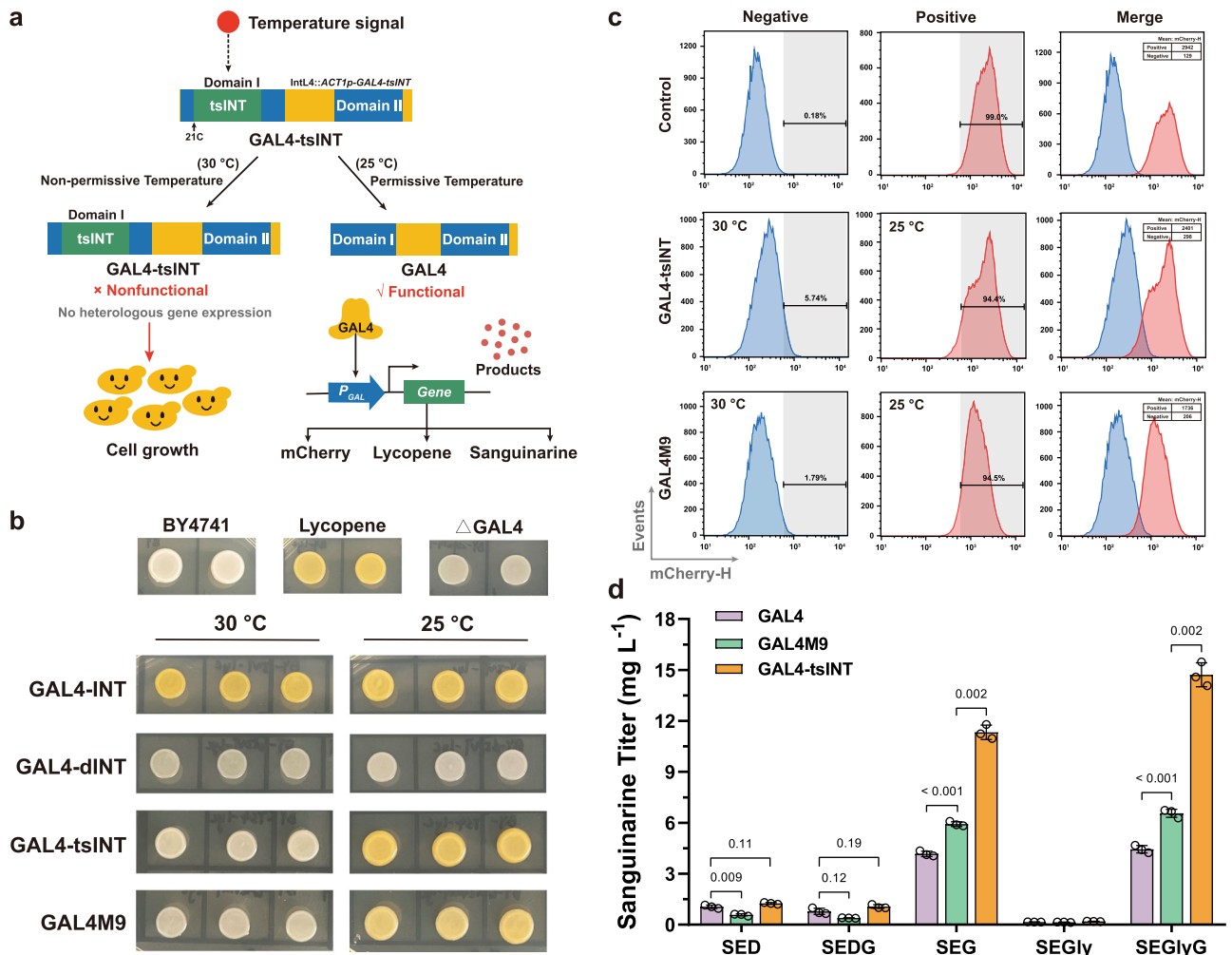

**Fig. 1 | Design and development of SIMTeGES. a** Temperature-dependent splicing intein variants are inserted into the GAL4 DNA Binding Domain. At the non-permissive temperature (30 °C), intein remains unspliced, preventing GAL4-tsINT from binding to a specific DNA region and activating downstream GAL promoter expression, thereby directing available resources for cell growth. Upon reaching a specific biomass, transitioning to the permissive temperature (25 °C) induces intein self-splicing, resulting in the generation of functional GAL4 to initiate downstream gene expression and the transition from cell growth to product synthesis. **b** The colored metabolite lycopene serves as a reporter system, reflecting GAL4 activity for characterizing appropriate intein variants. **c** *mCherry* is employed as a reporter gene, utilizing flow cytometry to further investigate the activity and leaky expression of SIMTeGES. **d** SIMTeGES-GAL4 is employed under diverse carbon source conditions for stable production of sanguinarine. SED, SEDG, SEG, SEGly, and SEGlyG denote different fermentation conditions in SE medium, with the following carbon sources: 20 g L$^{-1}$ glucose, a combination of 20 g L$^{-1}$ glucose and 10 g L$^{-1}$ galactose, 20 g L$^{-1}$ galactose, 20 g L$^{-1}$ glycerol, and a combination of 10 g L$^{-1}$ glycerol and 20 g L$^{-1}$ galactose, respectively. Significance was calculated using two-way ANOVA followed by Tukey's multiple comparisons test. Data are presented as mean ± s.d. (*n* = 3 biologically independent samples). Source data are provided as a Source Data file.

Unfortunately, the activation capacity of GAL4M9 was sacrificed and might hinder the production of plant secondary metabolites with intricate and lengthy biosynthetic pathways (e.g., sanguinarine)[26]. Hence, we further investigated the transcriptional activity and dynamic regulation properties of the temperature-responsive systems (GAL4-tsINT and GAL4M9) using *mCherry* under the control of the *GAL1* promoter as a reporter gene. We measured the fluorescence signals in galactose media at 25 °C and 30 °C, respectively, using fluorescence microplate reader and flow cytometry. After 36 h, both GAL4-tsINT and GAL4M9 showed temperature-sensitive activity when compared with GAL4 and GAL4-dINT positive and negative controls (Fig. 1c and Supplementary Fig. 5). GAL4-tsINT exhibited stronger fluorescence intensity than GAL4M9 at 25 °C, indicating higher transcriptional activation capacity. At 30 °C, both systems showed low yet comparable levels of leakage expression, with GAL4-tsINT showing an activation dynamic around 14.1-fold.

Moreover, we evaluated the temperature-responsive kinetics of GAL4-tsINT and GAL4M9. After 24 h incubation, we shifted the temperature from 30 °C to 25 °C, and collected samples at 6 h intervals for analysis. As shown in Supplementary Fig. 6, GAL4-tsINT and GAL4M9 showed similar temperature-sensitive kinetic properties, whose expression was triggered at 6 h and gradually slowed down at 24 h, suggesting the most robust temperature-controlled transcriptional activation between 6 h and 24 h. Compared with GAL4M9, GAL4-tsINT consistently exhibited more pronounced changes in fluorescence intensity, highlighting that SIMTeGES preserves the maximal transcriptional activation capacity of GAL4 while displaying decent temperature sensitivity.

Afterward, we employed SIMTeGES for the biosynthesis of lycopene to further compare GAL4-tsINT and GAL4M9. Given the inhibitory effect of glucose on the GAL system, we investigated the effect of various carbon sources on the performance of the temperature-responsive systems and, accordingly, the production of lycopene. Under all conditions, GAL4-tsINT consistently outperformed GAL4M9 (Supplementary Fig. 7), particularly when both glycerol and galactose were present (YPGlyG), with the lycopene titer of GAL4-tsINT 15.5-fold

higher than that of GAL4M9. Compared with the *mCherry* reporter system with only one gene, the lycopene biosynthesis pathway involving four steps of enzyme catalysis likely accentuated the disparities in transcriptional activity between GAL4-tsINT and GAL4M9. Surprisingly, GAL4-tsINT failed to exhibit significant advantages in the presence of glucose, probably due to the interplay of multiple signals, including the glucose repression signal and temperature-dependent activation signal. Nevertheless, the lycopene titer with GAL4-tsINT was still 3.7-fold higher than that with GAL4 under YPGlyG culture conditions. Consequently, we successfully reset the decoupling signal of the GAL system from high-concentration glucose to temperature. SIM-TeGES not only maintained high expression levels but also established a more rigorous and easily controllable growth and production decoupling system, making it more suitable for the biosynthesis of complex plant secondary metabolites (Supplementary Table 1).

## Mechanism exploration of SIMTeGES

The intein splicing mechanism is a well-defined process, delineated into four principal steps[33] (Supplementary Fig. 8). Typically, the initial step is considered the key limiting step, where the S/O atom of the *N*-terminal cysteine or threonine of the inteins attacked the α-carbon of the preceding amino acid, resulting in N-O or N-S acyl rearrangement[38,39]. It is worth noting that the conformational strain on the backbone of the precursor protein is a vital indicator of this process. Quantifying these primary chain distortions involves assessing the deviation from peptide bond planarity (ω value) near the scissile peptide bond (with 180° as the ideal value), causing an energy increase of over 5 kcal/mol for each distorted residue[39] (Fig. 2a). Therefore, we employed AlphaFold 2.0 and MD simulations to model GAL4-tsINT and adjusted the structures at both 30 °C and 25 °C to further explore the

mechanism of conditional splicing by analyzing their conformational changes.

After molecular dynamic equilibrium (Supplementary Fig. 9), we computed the average planarity of the peptide bond to be 176.0°, formed by residues C21 and F22 within the GAL4-tsINT precursor protein at 30 °C during the 40 to 50 ns (Fig. 2b), closely mirroring the ideal value of 180°. This indicated minimal or negligible distortion in the backbone at 30 °C, insufficient to trigger N−O or N−S acyl rearrangement. As a result, the intein within the precursor protein remained unspliced, rendering it devoid of transcriptional activation capacity. Conversely, at 25 °C, the average planarity of the peptide bond at the cleavage site in the precursor protein was determined to be 166.7°, significantly deviating from the ideal value. This pronounced conformational strain at 25 °C facilitated the nucleophilic attack by the S/O atom of cysteine or threonine, thereby enabling intein self-hydrolysis for the generation of functional target proteins.

## Exploration of general applicability of SIMTeGES

Upon our understanding of temperature-sensitive intein splicing mechanism, we further evaluated the performance of SIMTeGES for temperature-controlled expression of more proteins (e.g., GAL80 and mCherry) in various hosts (e.g., *P. pastoris* and mammalian cells). First, we inserted INT, dINT, and tsINT at position C277 of the GAL80 protein (SIMTeGES-GAL80), whose encoding genes were integrated into the lycopene-producing strain (Supplementary Fig. 10). Given the role of GAL80 in repressing GAL4 transcriptional activation, colonies remained colorless when GAL80 was active but turned to the carotenogenic color upon GAL80 inactivation. Our results showed that GAL80-INT led to a modest decrease in the inhibitory activity of GAL80, as colonies with a pale-yellow color (Fig. 2c). Notably, GAL80-

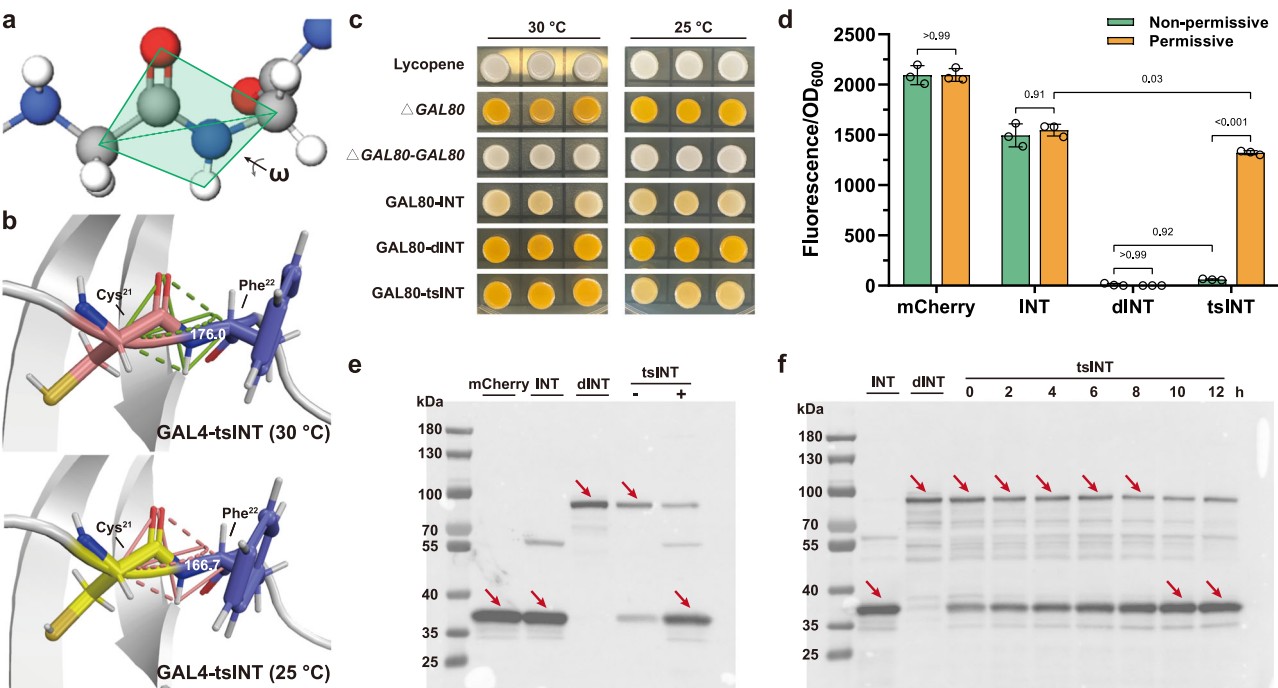

**Fig. 2 | Investigation of the molecular mechanism and general applicability of SIMTeGES. a** Quantification of conformational strain on the backbone involves assessing the deviation from the peptide bond planarity near the scissile peptide bond. Gray spheres represent C atoms, blue spheres represent N atoms, and red spheres represent O atoms. **b** Measurement of the average planarity of the peptide bond in GAL4-tsINT after achieving molecular dynamic equilibrium, recorded at both 30 °C and 25 °C. **c** SIMTeGES for temperature-controlled expression of GAL80, employing the colored metabolite lycopene as a reporter system. **d** SIMTeGES for temperature-dependent expression of mCherry. **e** Western blot

analysis of mCherry-INT, mCherry-dINT, and mCherry-tsINT, validating the direct correlation between host protein activity and intein splicing. **f** Western blot analysis of the splicing kinetics of mCherry-tsINT. The unspliced protein is ~85 kDa, while the spliced form is around 35 kDa. Blots were representative of at least two independent experiments (2e and 2 f). Significance was calculated using two-way ANOVA followed by Tukey's multiple comparisons test. Data are presented as mean ± s.d. (*n* = 3 biologically independent samples). Source data are provided as a Source Data file.

tsINT exhibited robust temperature-dependent behavior: under non-permissive temperature, it became nonfunctional and resembled the carotenogenic color of the *gal80* disrupted strain colonies; under permissive temperature, its performance closely mirrored GAL80-INT, preserving most of the inhibitory activity of GAL80.

In addition to transcriptional factors (e.g., GAL4 and GAL80), we further evaluated its application in temperature-controlled expression of target proteins directly, with mCherry chosen as a reporter protein (SIMTeGES-mCherry, Supplementary Fig. 10). As shown in Fig. 2d and Supplementary Fig. 11, we successfully achieved temperature-regulated and minimal leakage expression of mCherry-tsINT under non-permissive conditions (~5.21% determined by flow cytometry), while the positivity rate under permissive conditions reached 99.8%, further demonstrating strong activity and low leakage expression levels of SIMTeGES. Furthermore, we validated SIMTeGES in different host cells, with a temperature-controlled expression of mCherry achieved in both *P. pastoris* and mammalian cells (Supplementary Fig. 12 and 13). These results underscored the versatility of SIMTeGES for multiple proteins across diverse species.

Consistent with our genetic analysis, we verified intein splicing at the protein level. Specifically, mCherry-INT was predominantly detected in its spliced form (~35 kDa), showcasing the efficient self-splicing capability of inteins (Fig. 2e). Conversely, mCherry-dINT existed as an unspliced form, around 85 kDa. The temperature-dependent mCherry-tsINT exhibited an unspliced form like mCherry-dINT under non-permissive conditions yet aligned with mCherry-INT in a spliced form under permissive conditions. These findings further validated the direct correlation between protein activity and intein splicing. Moreover, the ratio of spliced to unspliced proteins in mCherry-tsINT at various time points provides an intuitive insight into the splicing kinetics of the temperature-sensitive intein. During the transition from non-permissive to permissive conditions, we observe a gradual decrease in the proportion of unspliced proteins over time, accompanied by an increase in the proportion of spliced proteins (Fig. 2f), with most proteins present in the spliced form after 8 h.

## Stable production of sanguinarine using SIMTeGES

Initially, considering the involvement of 6 cytochrome P450 enzymes in the sanguinarine biosynthesis pathway (Fig. 3), we implemented several strategies to optimize the microenvironment for the expression and activity of P450s (Supplementary Fig. 14). We constructed strain SAN219 by overexpressing *ZWF1, GAPN, ICE2, INO2*, and *EcCFS* in the previously reported sanguinarine yeast strain SAN006[19]. Increasing the expression of glucose-6-phosphate dehydrogenase[40] (ZWF1) and introducing glyceraldehyde-3-phosphate dehydrogenase[41] (GAPN) from *Streptococcus mutans* are general approaches to enhance the availability of NADPH, a crucial cofactor for electron transfer in the CYP system. Moreover, the INO2/INO4 transcription factor complex activates phospholipid biosynthesis, and the transmembrane protein ICE2 stabilizes cytochrome P450 reductase (CPR) and other membrane proteins in the endoplasmic reticulum (ER)[42,43]. Unfortunately, none of these genome modifications resulted in a significant increase in sanguinarine production (Supplementary Fig. 15). Meanwhile, knockout out of *GAL80* in the strain SAN220 resulted in the loss of sanguinarine synthesis capability during passaging. We speculated that leakage expression of the conventional GAL regulon under low glucose concentrations led to an inadequate decoupling of cell growth and product synthesis, which in turn exerted heavy metabolic burdens and even cytotoxicity on the yeast strain. Thus, we propose SIMTeGES as a promising and effective strategy for alleviating the bottleneck in the development of high-yield sanguinarine-producing strains.

Consequently, we constructed two strains, SAN220-tsINT and SAN220-M9, by integrating two temperature-responsive GAL systems into the sanguinarine-producing strain SAN219. In the presence of galactose, both GAL4M9 and GAL4-tsINT exhibited significantly improved phenotypes compared with GAL4. Especially when a mixed carbon source of galactose and glycerol was used, GAL4-tsINT achieved a sanguinarine titer of up to 14.71 mg L$^{-1}$, surpassing GAL4 by 3.31-fold and GAL4M9 by 2.24-fold, respectively (Fig. 1d). In contrast to the lycopene-producing strain, all strains showed low sanguinarine titers when cultivated with glucose. We hypothesized that glucose, serving as a repressive carbon source, exerts a post-inhibitory effect on GAL promoters, especially in the intricate and lengthy sanguinarine biosynthetic pathway. Additionally, the mutual interference between the glucose repression signal and the temperature-controlled activation signal led to suboptimal performance. SIMTeGES effectively mitigated these limitations by replacing glucose with temperature as the decoupling signal without sacrificing heterologous gene expression levels. After subjecting the yeast strain SAN220-tsINT to five consecutive passages, we observed relatively stable sanguinarine titers, indicating the alleviation of the strain stability challenge caused by sanguinarine cytotoxicity (Supplementary Fig. 16).

Subsequently, we performed a comprehensive transcriptome analysis focusing on the cluster of genes involved in central carbon metabolism and the heterologous pathway to investigate the underlying reasons for significant variations in sanguinarine production with different carbon sources (Supplementary Fig. 17). As anticipated, when glycerol and galactose were used as mixed carbon sources, the expression levels of most heterologous genes and galactose assimilation associated genes were significantly increased. Although some genes involved in glycolysis and the pentose phosphate pathway (PPP) were downregulated, genes associated with the tricarboxylic acid (TCA) cycle displayed upregulation, thereby enhancing cellular robustness. Additionally, we identified several upregulated genes, including *FBP1* (encoding fructose-1,6-bisphosphatase), *RKI1* (encoding ribose-5-phosphate ketol-isomerase) and *TPI1* (encoding triose phosphate isomerase), which played crucial roles in gluconeogenesis and the non-oxidative stages of the PPP, facilitating the assimilation of galactose and enhancing the supply of erythrose-4-phosphate (E4P). These findings further supported the beneficial effects of using glycerol and galactose as mixed carbon sources in the GAL system, where temperature acted as the sole decoupling signal.

## Identification and functional expression of the flavoprotein berberine bridge enzyme

Reticuline oxidases belong to the berberine bridge enzyme family (BBE) and are responsible for the stereospecific conversion of (*S*)-reticuline to (*S*)-scoulerine (Fig. 4a). Our previous work has shown that increasing reticuline titer did not significantly improve the yield of the final product sanguinarine[19], indicating that BBE might be the key rate-limiting step in the pathway. In vivo, experiments have confirmed that *Ps*BBE and *Ec*BBE, derived from *Papaver somniferum* and *Eschscholzia californica*, exhibited reticuline oxidase activity[44,45]. To explore the catalytic potential of different reticuline oxidase enzymes, we constructed a sequence similarity network (SSN) for the BBE family (Pfam08031) (Supplementary Fig. 18). SSN allows for the visualization of protein sequence relatedness and clustering based on similarity thresholds[46]. By setting the alignment score threshold (AST) to 94 and restricting sequence lengths to 450-660 amino acids, we identified 8,812 sequences containing 15 different SwissProt descriptions. The BBE-likes, reticuline oxidases, and tetrahydroberberine oxidases all clustered together in SSN (Fig. 4b). From this cluster, we selected *Cj*BBE from *Coptis japonica* and *Mc*BBE from *M. cordata* for functional characterization.

*Ps*BBE has been observed to be expressed in plant compartment vesicles with a 23-amino acid signal peptide[47]. According to the Philius prediction server from the Yeast resource center[48], all BBEs were predicted to be non-cytoplasmic proteins. Considering the variation of signal peptide processing and localization in a heterologous host,

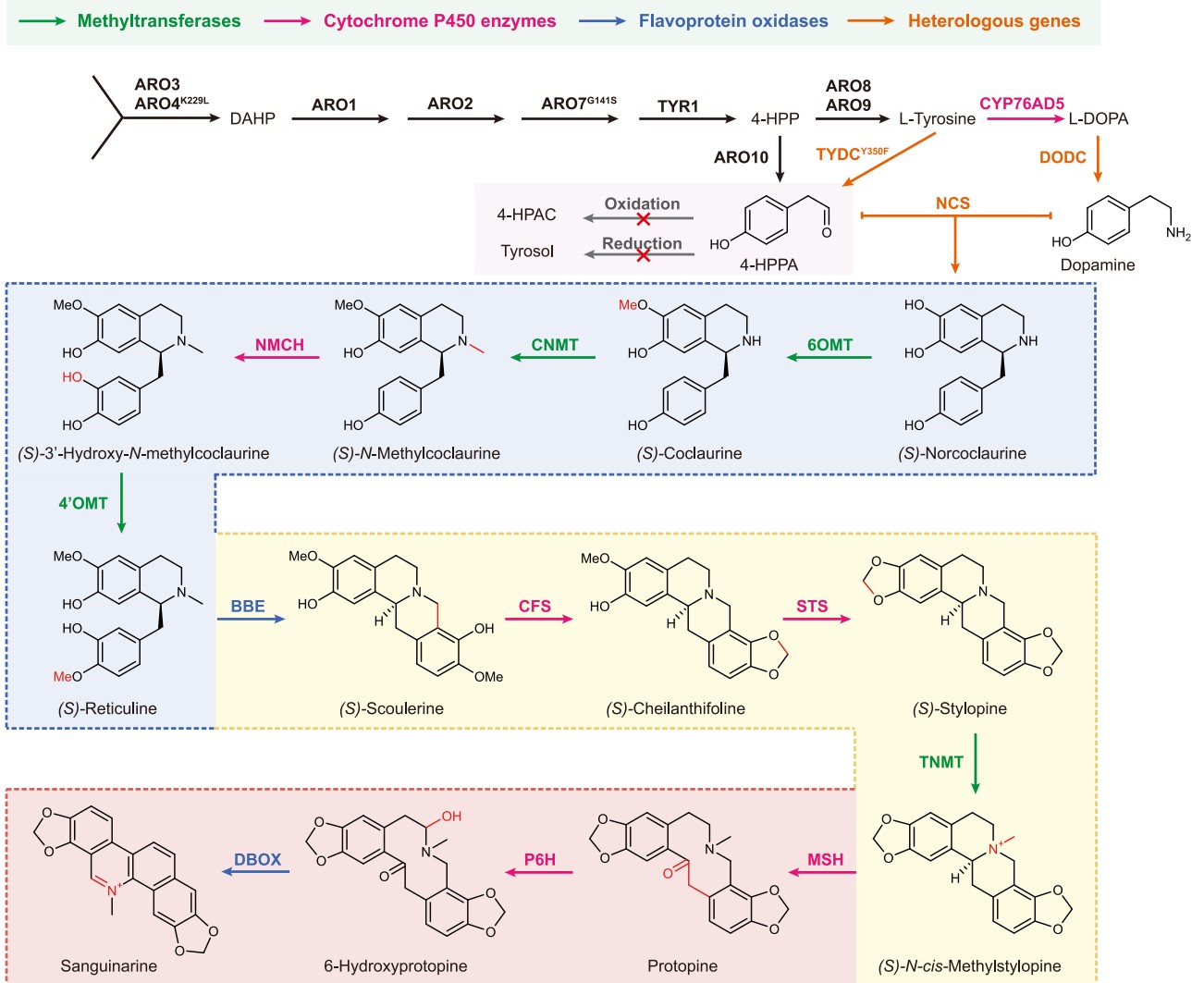

**Fig. 3 | Complete biosynthetic pathway for the production of sanguinarine in yeast.** Endogenous genes are depicted in black, while heterologous genes are shown in orange. Within the pathway, SAM-dependent methyltransferases, cytochrome P450 enzymes, and flavoprotein oxidases are highlighted in green, magenta, and blue, respectively. The pathway starts from the shikimate pathway intermediate, tyrosine, and involves the introduction of a total of 17 heterologous genes, including *CPR1* and *CYB5*, to achieve de novo biosynthesis of sanguinarine. Abbreviations not already defined in the text are as follows: 4-HPP, 4-hydroxyphenylpyruvate; L-DOPA, L-3,4-dihydroxyphenylalanine; CYP76AD5, tyrosine hydroxylase; DODC, DOPA decarboxylase; TYDC, tyrosine decarboxylase; NCS, norcoclaurine synthase; 6OMT, norcoclaurine 6-*O*-methyltransferase; CNMT, coclaurine *N*-methyltransferase; NMCH, *N*-methylcoclaurine 3'-hydroxylase isozyme 2; 4'OMT, 3'-hydroxy-*N*-methyl-coclaurine 4'-*O*-methyltransferase 2; BBE, reticuline oxidase; CFS, cheilanthifoline synthase; STS, stylopine synthase; TNMT, tetrahydroprotoberberine *N*-methyltransferase; MSH, methyltetrahydroprotoberberine 14-monooxygenase; P6H, protopine hydroxylase; DBOX, dihydrobenzophenanthridine oxidase.

we truncated the signal peptide of BBEs from different species (Supplementary Fig. 19) and fused an MBP (maltose binding protein) tag to their *N*-terminus to facilitate functional expression in the cytoplasm. To evaluate the activity of the BBE variants, we constructed the reticuline accumulation strain RET202, by knocking out *PsBBE*, *EcCFS*, and *EcSTS* from the sanguinarine-producing strain SAN006. Then, we evaluated and compared four BBEs (*PsBBE*, *EcBBE*, *CjBBE*, and *McBBE*), their signal peptide-truncated variants (t*Ps*BBE, t*Ec*BBE, t*Cj*BBE, and t*Mc*BBE), and variants with the signal peptide replaced with an MBP tag (MBP-t*Ps*BBE, MBP-t*Ec*BBE, MBP-t*Cj*BBE, and MBP-t*Mc*BBE). Our results indicated that *Mc*BBE exhibited high reticuline oxidase activity, and the MBP-t*Mc*BBE variant had a higher scoulerine conversion rate (~87% compared to a conversion rate of ~23% for the most commonly used *Ps*BBE) (Fig. 4c). Interestingly, all MBP fusion variants showed improved conversion rates. Noteworthy, while *Cj*BBE and t*Cj*BBE exhibited almost no reticuline oxidase

activity, MBP-t*Cj*BBE had a conversion rate of ~22%, highlighting the significance of MBP tag on the functional expression of BBEs in yeast.

To ascertain the localization of BBEs in yeast and evaluate the effect of signal peptide and MBP fusion tag on protein expression and cellular localization, we fused EGFP to the *C*-terminus of *Cj*BBE, *Mc*BBE, and their variants. While the expression of BBEs and tBBEs was too low to determine their localization using confocal microscopy, MBP-tBBEs were clearly expressed in the cytoplasm (Fig. 4d). The fluorescence intensity significantly increased after signal peptide truncation and MBP fusion, indicating a positive effect of MBP on soluble expression of BBEs. Consequently, integrating *MBP-tMcBBE* into the genome of SAN220-tsINT resulted in the construction of SAN221, which significantly decreased reticuline accumulation and accordingly increased the production of sanguinarine by 1.53-fold to a titer of 21.05 mg L$^{-1}$ (Fig. 4e).

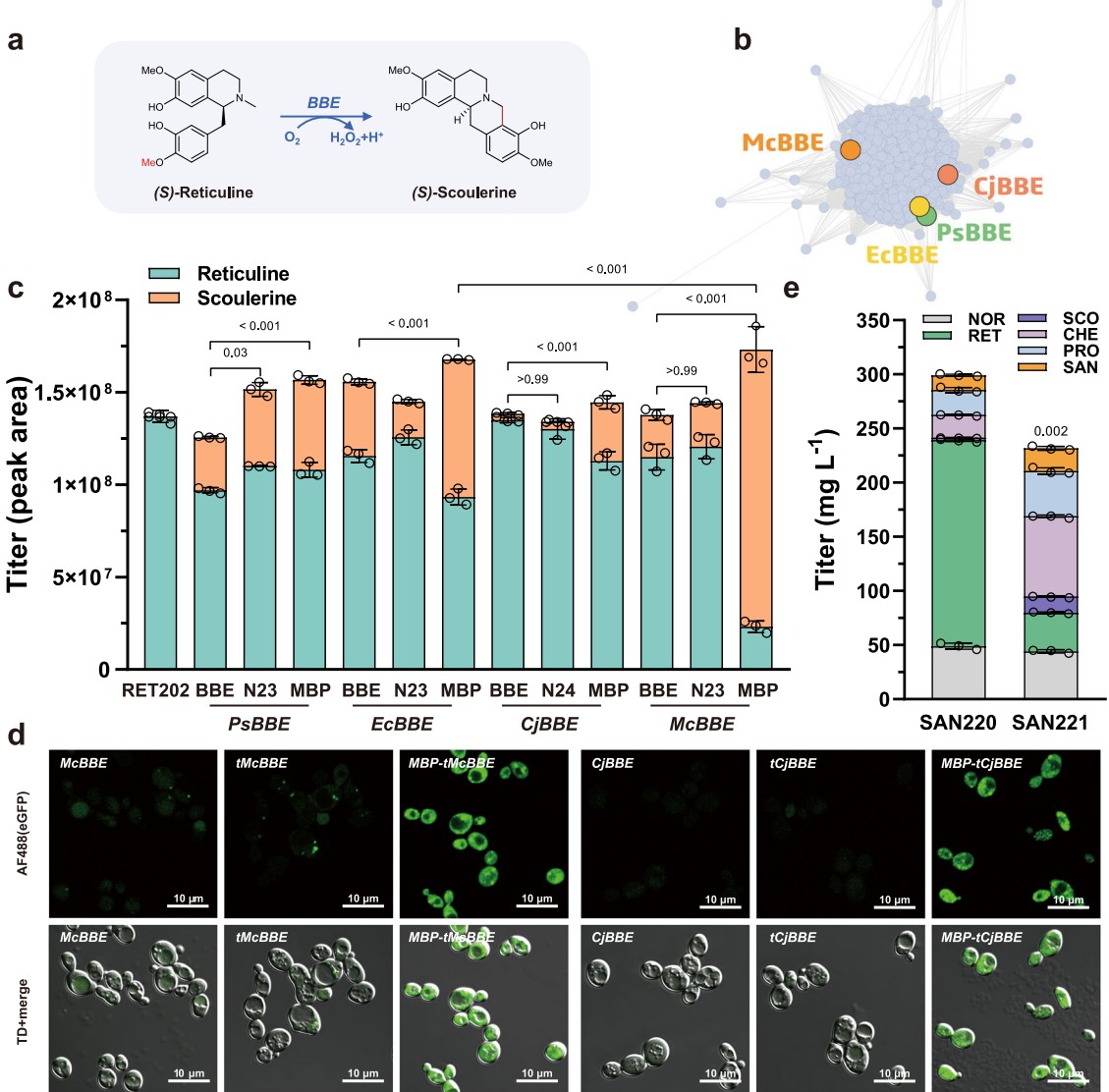

**Fig. 4 | Identification and functional expression of the flavoprotein berberine bridge enzyme (BBE). a** Enzymatic reaction catalyzed by reticuline oxidases. **b** The clustering of reticuline oxidases in SSN of the BBE family, leading to the identification of two BBEs derived from *C. japonica* and *M. cordata*. **c** Characterization of BBE variants from different species and those modified with signal peptide truncation and/or an *N*-terminal MBP fusion tag in the RET202 strain. The significance of scoulerine titer differences was calculated using two-way ANOVA followed by Tukey's multiple comparisons test. **d** Subcellular localization and expression of *Cj*BBE, *Mc*BBE, and their variants by fusing EGFP at the *C*-terminus, visualized

through fluorescence confocal microscopy. Micrographs were representative of at least two independent experiments. **e** Intermediate and product accumulation upon integration of *MBP-tMcBBE* in the SAN220-tsINT strain. Abbreviations not previously defined in the text are as follows: NOR, (*S*)-norcoclaurine; RET, (*S*)-reticuline; SCO, (*S*)-scoulerine; CHE, (*S*)-cheilanthifoline; PRO, protopine; SAN, sanguinarine. The significance of sanguinarine titer differences was calculated using two-way ANOVA followed by Sidak's multiple comparisons test. Data are presented as mean ± s.d. (*n* = 3 biologically independent samples). Source data are provided as a Source Data file.

## Transmembrane domain engineering for proper expression and localization of protopine 6-hydroxylase

The final cytochrome P450 enzyme (protopine 6-hydroxylase, P6H) in the sanguinarine pathway catalyzes the 6-hydroxylation of protopine, followed by non-enzymatic intramolecular rearrangement to form the benzophenanthridine scaffold of dihydrosanguinarine[36,49]. Analysis of intermediate metabolites in strain SAN221 revealed significant accumulation of protopine, suggesting that *Mc*P6H from *M. cordata* is one of the major bottlenecks (Fig. 4e). Previous studies have reported that *Mc*P6H exhibited higher catalytic activity than *Ec*P6H from *E. californica*, but its protein expression level was much lower in yeast[8]. Thus, we attempted to engineer the *N*-terminal α-helix to achieve proper localization and functional expression of *Mc*P6H (Fig. 5a).

Through sequence alignment between *Ec*P6H and *Mc*P6H, a non-aligning region of 14 consecutive serine residues KKSSSSSSSSSSSSSS was identified in *Mc*P6H (Supplementary Fig. 20a). We used AlphaFold 2.0 to predict the structure of *Mc*P6H and obtained a low confidence score for this sequence (Supplementary Fig. 20b), indicating that this sequence might not contribute to protein activity but impair protein stability. Thus, we removed this sequence to create a *Mc*P6H variant (*Mc*P6Hs). We speculated that correct protein expression and localization might be more critical than catalytic activity in the reconstruction of heterologous biosynthetic pathways. Thus, we designed two chimeric proteins with engineered transmembrane domains: *Ec*CFS[1-83]-*Mc*P6Hs[84-522] and *Ec*P6H[1-39]-*Mc*P6Hs[33-522].

To evaluate the activity of P6H variants, we knocked out *Mc*P6H in strain SAN221, resulting in the construction of the protopine-

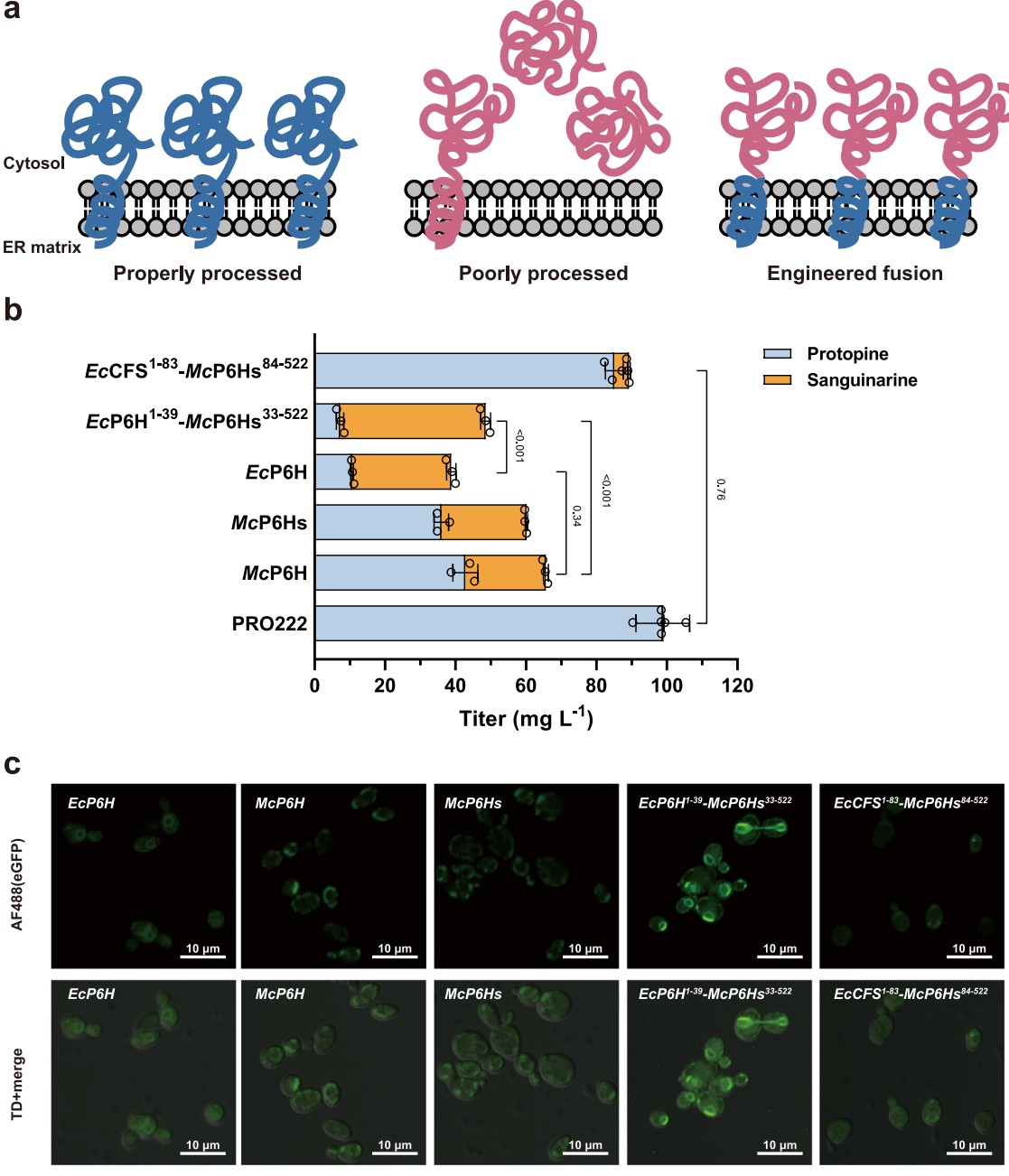

**Fig. 5 | Transmembrane domain engineering for proper expression and localization of protopine 6-hydroxylase (P6H). a** Schematic of the chimeric P6H engineering strategy to address improper processing and localization in yeast. **b** Characterization of the activity of P6H variants from various species and engineered fusions in the PRO222 strain. **c** Investigation of the subcellular localization and expression of P6H variants by fusing EGFP at the *C*-terminus, visualized using fluorescence confocal microscopy. Micrographs were representative of at least two independent experiments. The significance of sanguinarine titer differences was calculated using two-way ANOVA followed by Sidak's multiple comparisons test. Data are presented as mean ± s.d. (*n* = 3 biologically independent samples). Source data are provided as a Source Data file.

producing strain PRO222. We then introduced *Mc*P6H, *Mc*P6Hs, *Ec*P6H, *Ec*CFS[1–83]-*Mc*P6Hs[84-522], and *Ec*P6H[1–39]-*Mc*P6Hs[33-522] into PRO222 and evaluated the conversion of protopine to sanguinarine. Compared with the 35% conversion rate of *Mc*P6H, *Mc*P6Hs only showed a slight increase in sanguinarine production. In contrast, *Ec*P6H exhibited much higher activity in yeast, significantly increasing the conversion rate to 73% (Fig. 5b). For the two chimeric proteins, *Ec*CFS[1–83]-*Mc*P6Hs[84-522] showed a relatively low activity with a conversion rate of only 4.7%, while *Ec*P6H[1–39]-*Mc*P6Hs[33-522] demonstrated optimal catalytic activity, almost eliminating the accumulation of protopine and increasing the titer of sanguinarine to 41.31 mg L[−1] with a conversion rate of 85%. This

suggested that the transmembrane domain of *Ec*P6H was more compatible and significantly improved the localization and functional expression of *Mc*P6Hs in yeast.

The results of confocal fluorescence microscopy confirmed our hypothesis (Fig. 5c). *Ec*P6H localized on ER with a circular distribution in yeast cells. In contrast, *Mc*P6H and *Mc*P6Hs accumulated at the edges of the cells, indicating incorrect localization, which impaired their catalytic activities. While the cellular localization of *Ec*CFS[1–83]-*Mc*P6Hs[84-522] could not be determined due to low expression level, *Ec*P6H[1–39]-*Mc*P6Hs[33-522] displayed a distribution similar to *Ec*P6H and localized correctly on ER. Notably, the fluorescence intensity of

*Ec*P6H[1–39]-*Mc*P6Hs[33-522] was significantly higher than that of *Ec*P6H, demonstrating that fusion of the transmembrane domain of *Ec*P6H and *Mc*P6Hs indeed facilitated protein expression and correct localization. The transmembrane domain engineering provides a useful example for enhancing the in vivo activity of cytochrome P450 enzymes in yeast.

## Comprehensive pathway optimization

After addressing two key rate-limiting steps in the pathway, our focus shifted to enhancing precursor supply, cofactor engineering, and cellular detoxification capacity to redirect metabolic flux towards sanguinarine production. PEP derived from glycolysis and E4P derived from PPP serve as important precursors in the upstream pathway. Phosphoketolase (XFPK) from *Leuconostoc mesenteroides* has been shown to enable the conversion of xylose-5-phosphate (X5P) and/or fructose-6-phosphate (F6P) into acetyl phosphate and glyceraldehyde-3-phosphate (GAP)/E4P, thereby increasing the availability of E4P[40]. Additionally, the enhancement of the shikimic acid pathway through *Escherichia coli* shikimate kinase II (*Eco*AROL) and the deletion of aldehyde reductase (ARI1) have been reported to minimize the reduction of 4-HPAA to tyrosol[28]. Consequently, based

on SAN223-4, we introduced *Lm*XFPK and *Eco*AROL while knocking out the redundant oxidoreductase *ARI1* to construct SAN224, resulting in a sanguinarine titer of 46.78 mg L[−1] with the accumulation of norcoclaurine and reticuline increased by 1.55-fold and 1.37-fold, respectively (Fig. 6a). To optimize the microenvironment of the four SAM-dependent methyltransferases, replenishing methionine synthesis is crucial for establishing a complete SAM cycle (Supplementary Fig. 21). By employing the *ACT1* promoter to alleviate feedback inhibition of the *MET17* promoter, the catalytic activity of SAM-dependent methyltransferases (6OMT, CNMT, 4'OMT, and TNMT) in the pathway was further enhanced, leading to a 1.33-fold increase in reticuline accumulation, sanguinarine production in SAN225 was increased by 1.22-fold, reaching 57.22 mg L[−1]. Moreover, previous studies have demonstrated that sanguinarine exhibits antimicrobial activity by generating reactive oxygen species (ROS) within cells at high doses, leading to DNA damage and cell apoptosis[50]. Superoxide dismutase (SOD) plays a vital role in eliminating ROS by converting superoxide anions to hydrogen peroxide[51]. Overexpressing the endogenous *SOD1* in SAN226 reduced ROS levels (Supplementary Fig. 22), leading to a slight improvement in sanguinarine production to 61.30 mg L[−1].

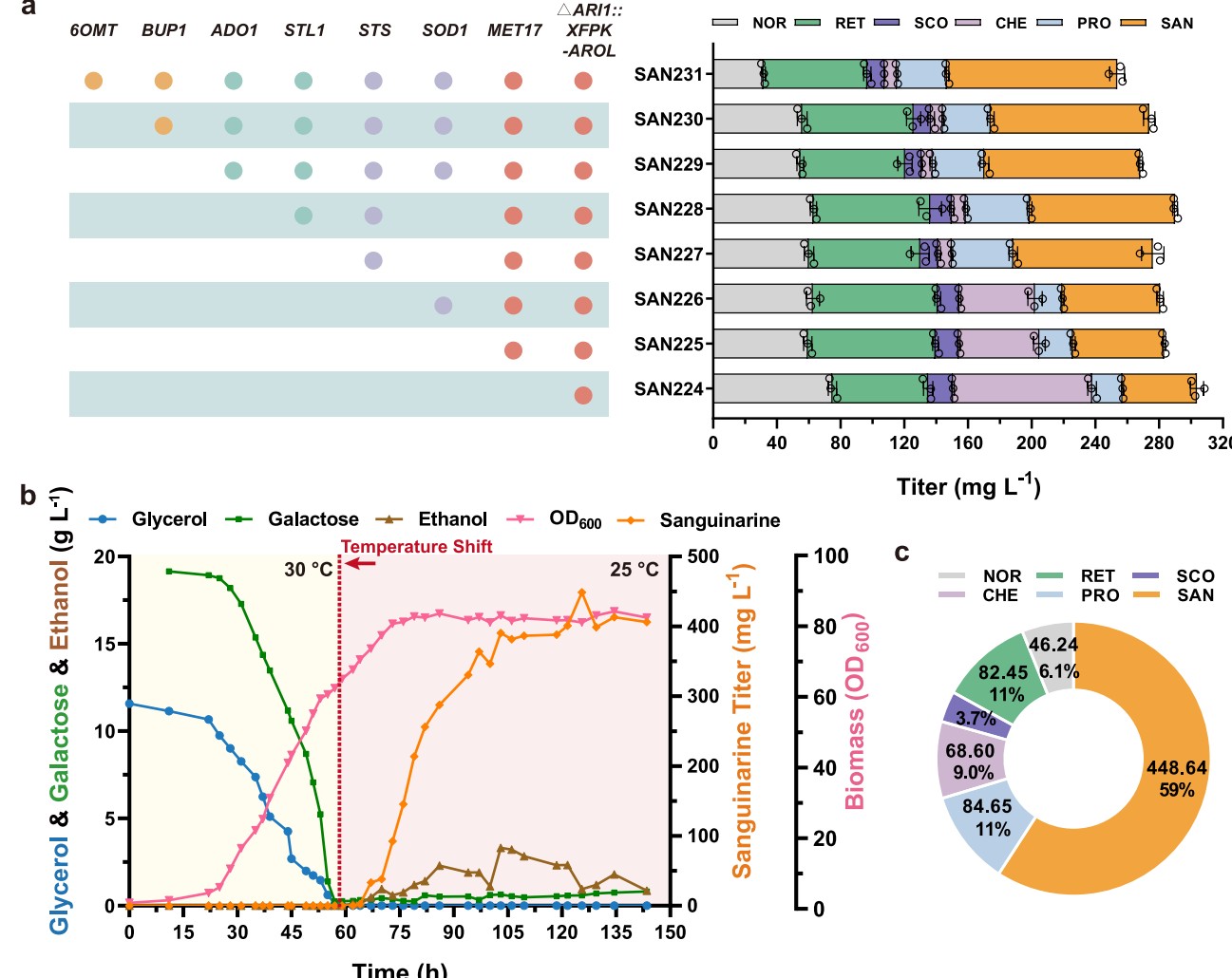

**Fig. 6 | Comprehensive pathway optimization and temperature-regulated fed-batch fermentation. a** Global optimization based on the SAN223 strain, showing the accumulation of various intermediates and products. **b** Profiles of cell growth, sanguinarine production, and carbon source consumption throughout the temperature-regulated fed-batch fermentation with SAN232. **c** Analysis of the accumulation of products and various intermediates resulting from SAN232 fermentation. Abbreviations not previously defined: NOR, (*S*)-norcoclaurine; RET, (*S*)-reticuline; SCO, (*S*)-scoulerine; CHE, (*S*)-cheilanthifoline; PRO, protopine; SAN, sanguinarine. Data are presented as mean ± s.d. (*n* = 3 biologically independent samples). Source data are provided as a Source Data file.

To further investigate the rate-limiting steps in the pathway, we employed qPCR to assess the expression levels of 16 heterologous genes in the SAN220-tsINT strain (Supplementary Fig. 23), among which the expression levels of *EcSTS* and *Ps6OMT* were significantly lower than the others. Therefore, we introduced an additional copy of the *EcSTS* expression cassette into SAN225, resulting in a 5.58-fold decrease in cheilanthifoline accumulation, a 2.23-fold increase in protopine accumulation, and a 1.53-fold increase in sanguinarine production (with a titer of 87.50 mg L$^{-1}$), respectively. Furthermore, we overexpressed the glycerol/H$^+$ symporter encoding gene *STL1*[52,53] and adenosine kinase gene *ADO1*[40] to improve glycerol utilization and replenish homocysteine in the SAM cycle. This slightly promoted sanguinarine synthesis, and as a result, SAN229 exhibited a sanguinarine titer of 97.94 mg L$^{-1}$. It has been reported that a class of BIA uptake permeases, known as BIA importers (BUP), localized in the *Opium poppy* plasma membrane, was able to transport various BIAs and certain pathway precursors (e.g., dopamine) and enhance the uptake rate of codeine and intermediates in yeast cells[54,55]. Unfortunately, the introduction of *BUP1* into the SAN229 strain failed to significantly increase sanguinarine production, possibly due to mislocalization of BUP1 in the yeast organelle (Supplementary Fig. 24). Finally, we engineered SAN231 with enhanced expression of *Ps6OMT*, resulting in a 1.77-fold decrease in norcoclaurine accumulation and the increase of sanguinarine production to 106.87 mg L$^{-1}$. We performed qPCR analysis on strains SAN223-4 and SAN231 to assess the abundance of each enzyme involved in heterologous expression or endogenous overexpression, thereby confirming the efficacy of our engineered modifications at the transcriptional level (Supplementary Fig. 25). We also constructed strain SAN334 by replacing GAL4-tsINT in strain SAN231 with the wild-type GAL4, resulting in a significant decrease in sanguinarine production, further validating the effectiveness of SIMTeGES (Supplementary Fig. 26).

For the temperature-regulated fed-batch fermentation, we constructed the prototrophic haploid strain SAN232 by complementing the auxotrophic markers in SAN231. In the early stages of fermentation, we maintained the temperature at 30 °C to support strain growth with hardly any sanguinarine production (Fig. 6b). When the strain entered the mid to late logarithmic growth phase at 59 h, we lowered the culture temperature to 25 °C to activate the temperature-sensitive GAL4-tsINT, thereby initiating the expression of heterologous genes. Notably, the relatively lower temperature has been found to favor the functionality of several plant-derived P450 enzymes[36]. After 70 h, while biomass was only slightly increased, sanguinarine started to accumulate to high levels. Finally, biomass reached an OD$_{600}$ of 84.32, and we achieved a sanguinarine titer of 448.64 mg L$^{-1}$ after 125.5 h of fermentation. Furthermore, different from flask fermentation, most intermediates were not accumulated to high levels in the fermentation broth, underscoring the high efficiency of sanguinarine biosynthesis (Fig. 6c).

### Efficient yeast cell factory for biosynthesis of halogenated BIA derivatives

Previously, we introduced a tyrosine decarboxylase variant[56], TyDC$^{Y350F}$, which converts L-tyrosine into 4-hydroxyphenylacetaldehyde (4-HPAA), a precursor of norcoclaurine synthase (NCS). Meanwhile, L-tyrosine can be transformed into L-3,4-dihydroxyphenylalanine (L-DOPA) by tyrosine hydroxylase (CYP76AD5), which is further utilized in the synthesis of L-dopamine, another precursor of NCS. By supplementing our high-yield sanguinarine cell factory with halogenated tyrosine (3-F-tyrosine, 3-Cl-tyrosine, and 3-I-tyrosine), we successfully synthesize diverse halogenated benzylisoquinoline derivatives. Our findings showed that feeding the cells with 0.5 mM 3-F-tyrosine resulted in the production of monofluorinated derivatives of all available pathway intermediates, including the production of fluorinated sanguinarine (Fig. 7a–e, Supplementary Fig. 27-31). As the derivatives could potentially be fluorinated at the 3' position catalyzed by TyDC$^{Y350F}$ or the 8 position catalyzed by CYP76AD5, we employed LC-MS/MS (QQQ) in production mode to extract the [M + H]$^+$ molecular weight of the halogenated derivatives and obtain their corresponding MS$^2$ spectra to determine the halogenation site[57]. For instance, the monofluorinated norcoclaurine had a [M + H]$^+$ molecular weight of 290.1, and the characteristic fragments at 125 indicated fluorination at the 3' position, while fragments at 179 indicated fluorination at the 8 position (Supplementary Fig. 27). The abundance ratio of 124.9 to 179.1 in the mass spectrum was 3.28, indicating that fluorination predominantly occurred at the 3' position of norcoclaurine, which could be due to the higher metabolite flux of the TyDC$^{Y350F}$ pathway when compared with CYP76AD5 or the steric hindrance at the 8 position. It is worth noting that we also detected a difluorinated derivative of norcoclaurine, which was less likely to undergo further enzymatic catalysis due to its increased molecular polarity. Due to the lack of structural references with characteristic fragments, we determined the fluorinated sanguinarine by comparing the MS$^2$ spectra before and after fluorination and identifying the major fragments with a mass difference of 18 (Fig. 7c).

When fed with 0.5 mM 3-Cl-tyrosine, we observed the production of chlorinated norcoclaurine and chlorinated reticuline, but not chlorinated scoulerine. Similarly, when supplemented with 0.5 mM 3-I-tyrosine, we only identified iodinated norcoclaurine. To further validate our findings, we employed high-resolution mass spectrometry (LC-MS-TOF)[27], facilitating the determination of precise molecular weights and isotopic distribution of major fragments, yielding consistent results (Fig. 7d, e, Supplementary Fig. 32–41, and Supplementary Table 2 and 3). These results suggest that variations in molecular size and polarity can influence subsequent catalysis. The biosynthesis of fluorinated sanguinarine highlights the potential of yeast as an expandable platform for constructing various benzylisoquinoline alkaloids.

## Discussion

Our work presents comprehensive metabolic engineering of yeast for the complete biosynthesis of sanguinarine. The final strain incorporates genome integration of 42 expression cassettes and disruption of 8 endogenous genes that strengthen the endogenous metabolic flux, overcome multiple bottlenecks in the biosynthesis pathway, and minimize the cytotoxicity of sanguinarine on yeast cells. The titer reaches up to 448.64 mg L$^{-1}$ for microbial cell factories with more than 15 plant-derived enzymatic reaction pathways, providing a classic example of the biosynthesis of plant secondary metabolites. Moreover, we establish SIMTeGES for temperature-controlled expression of *GAL4*, *GAL80*, and *mCherry* in *S. cerevisiae*, *P. pastoris*, and mammalian cells. The excellent temperature sensitivity and maximal protein activity properties make SIMTeGES an ideal system for biomanufacturing plant natural products by decoupling cell growth from product biosynthesis.

The identification and debottleneck of rate-limiting steps are essential for the construction of efficient microbial cell factories, particularly those with intricate and lengthy biosynthetic pathways[21]. Our study demonstrates a representative case study for systematic optimization of microbial cell factories at different levels, encompassing those at the enzyme, pathway, and global performance levels. At the transcriptional level, we can carry out qPCR or RNA-Seq analysis to pinpoint genes with relatively low expression levels, which can be addressed by increasing gene copy numbers[15]. At the protein level, it is recommended to perform fluorescence confocal microscopy first to determine the processing and localization of key enzymes of the plant secondary metabolite biosynthetic pathways[17]. *N*-terminal engineering (e.g., signal peptide truncation) has been shown to facilitate functional expression of target proteins in yeast. In the case of enzymes like BBEs with subcellular localization in plants, signal peptide truncation and MBP fusion expression have been found to enhance their solubility in

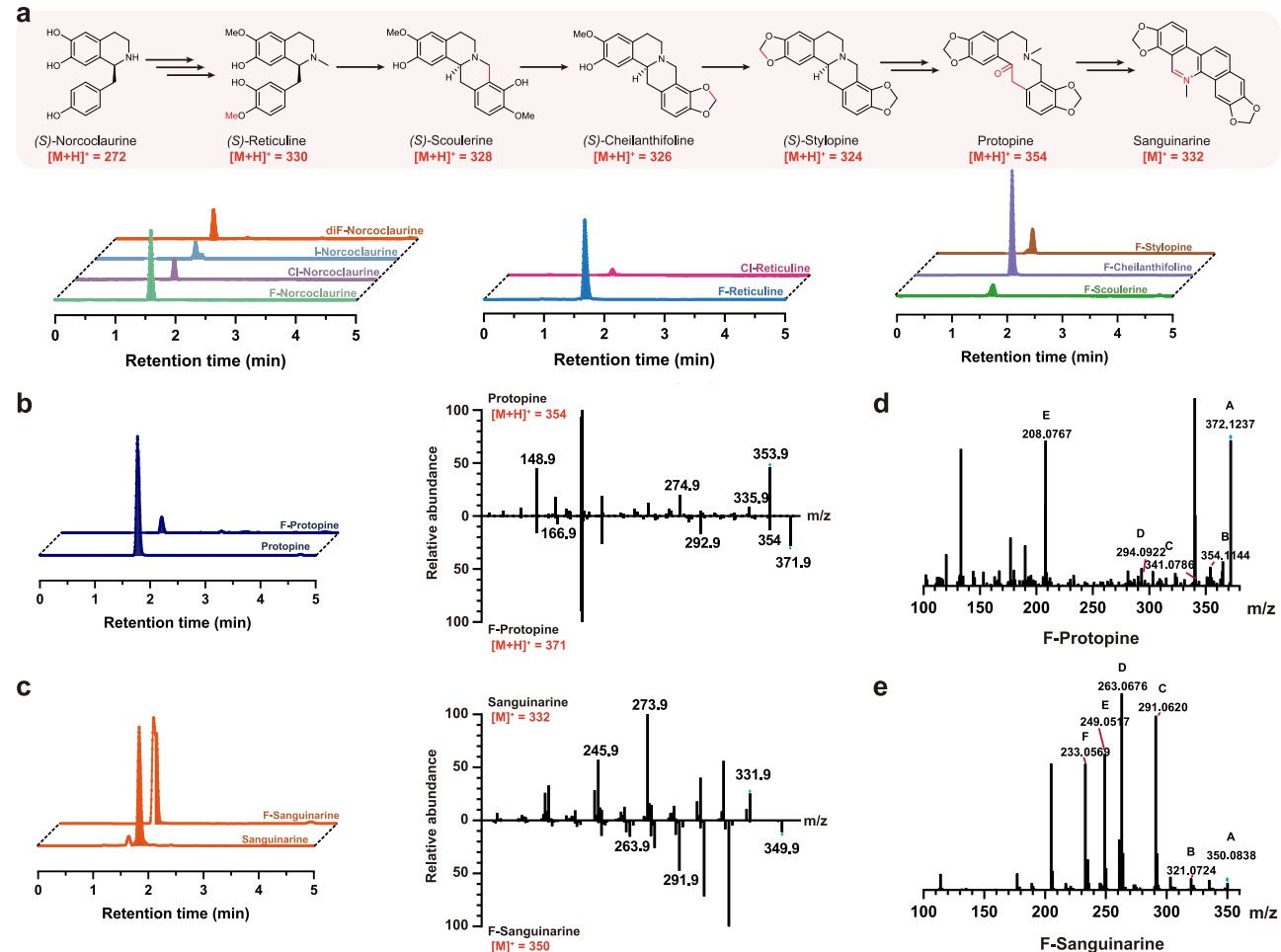

**Fig. 7 | Efficient yeast cell factory for biosynthesis of halogenated BIA derivatives. a** LC–QQQ traces displaying peaks of halogenated BIA derivatives after feeding 0.5 mM halogenated tyrosine. Specific MS² spectra detected through production mode and potential characteristic fragments are detailed in Supplementary Fig. 27-31. High-resolution MS² spectra captured by LC-MS-TOF are presented in Supplementary Fig. 32-41. **b** Comparison of LC–QQQ peaks and MS² spectra between protopine standard and F-protopine. **c** Comparison of LC–QQQ peaks and MS² spectra between sanguinarine standard and F-sanguinarine. **d** High-resolution MS spectra of F-protopine detected by LC-MS-TOF, with possible fragment formulas detailed in Supplementary Fig. 40. **e** High-resolution MS² spectra of F-sanguinarine by LC-MS-TOF, with possible fragment formulas detailed in Supplementary Fig. 41. Source data are provided as a Source Data file.

the cytoplasm of yeast. For heterologous expression of cytochrome P450s, the transmembrane domain engineering strategy used in our study is suggested to ensure correct *N*-terminal sorting of the nascent peptide and efficient ER localization[20]. At the metabolite level, detailed analysis of the biosynthesis intermediates and shunt byproducts is essential[12]. Most intermediates in the sanguinarine pathway are easily accessible, which facilitates the identification and debottlenecking of the rate-limiting steps of the biosynthetic pathway. At the metabolic network or systems level, the enhanced supply of precursor metabolites (e.g., E4P) and enzyme co-factors[40] (e.g., SAM for methyltransferases and NADPH for P450s), improved cell detoxification capacity and fitness (e.g., *SOD1* overexpression to decrease ROS level), and minimized product cytotoxicity (e.g., SIMTeGES to decouple cell growth from product biosynthesis) all contributed to the construction of a yeast cell factory for efficient sanguinarine production.

While this study has addressed several bottlenecks in sanguinarine biosynthesis, many aspects of the complex plant secondary metabolite pathway still require further investigation[14]. A recent study reported negative feedback inhibition of *Ps*4'OMT by reticuline, which should be addressed by protein engineering efforts to further enhance the metabolic fluxes towards sanguinarine biosynthesis[55]. The establishment of a fluorescence-based high-throughput screening platform will facilitate a more comprehensive and systematic exploration of

sanguinarine biosynthesis in yeast (Supplementary Fig. 42), including the alleviation of feedback inhibition and improvement of P450 activities via protein engineering, optimization of cellular microenvironment for pathway enzymes, along with enhanced sanguinarine export via transporter engineering, particularly when combined with machine learning[58,59] and BioFoundry[60,61] methodologies.

Halogenation significantly influences the bioavailability and activity of natural products, providing inspiration and reference for drug design. The introduction of the tyrosine decarboxylase variant, *Ps*TyDC^Y350F, which synthesizes 4-HPAA from tyrosine, not only demonstrates the divergence and convergence of the catalytic mechanism of plant aromatic amino acid decarboxylases, but also enriches the diversity of halogenated BIAs derivatives by incorporating halogenated tyrosine[56]. Among them, 3-fluorotyrosine is more readily accepted by the sanguinarine biosynthetic pathway due to its smaller steric hindrance. Similar results were reported in the integration of carbon-halogen bonds into alstonine (5 steps for halogenated alstonine biosynthesis from halogenated tryptophan)[31], berberine (9 steps for F-tetrahydrocolumbamine biosynthesis from 3-fluorotyrosine)[32], and noscapine (7 steps for halogenated reticuline biosynthesis from halogenated tyrosine)[27] biosynthetic pathways. However, as these derivatives are not native substrates for the pathway enzymes and exhibit lower conversion yield, particularly in longer pathways, our

findings leave some ambiguity regarding whether downstream enzymes (such as BBE) are incapable of tolerating chlorine substitution or there is an inadequate supply of the precursor Cl-reticuline, resulting in undetectable Cl-scoulerine. Moreover, we found that the proportion of 3'-halogenated compounds was much higher than that of 8-halogenated counterparts. When this manuscript was under review, another group also reported the same preference for the halogenation position[32]. In the present study, we report the detection of difluorinated norcoclaurine and fluorinated sanguinarine, demonstrating the biotransformation of halogenated derivatives through more than 15 biocatalytic steps (e.g., in the biosynthesis of F-sanguinarine from F-tyrosine). It is worth mentioning that most of the halogenated derivatives are not naturally occurring, highlighting the transformative impact of synthetic biology on natural product diversification and drug development.

In summary, we establish SIMTeGES for effectively decoupling cell growth from product synthesis. The temperature-dependent dynamic pathway control property and maintenance of high protein activities pave the way for the construction of microbial cell factories for high-yield production of a broader spectrum of natural products, particularly those with cytotoxicity. The demonstration of SIMTeGES for temperature-dependent expression of several proteins in different hosts underscores the potential applications. Additionally, the combination of SIMTeGES and systems metabolic engineering addresses major limitations in sanguinarine biosynthesis, with the final titer reaching up to 448.64 mg L$^{-1}$. The pioneering biosynthesis of halogenated sanguinarine opens horizons for the development of biogenic alkaloids with unique pharmacological properties.

## Methods

### Chemicals and reagents
Polyethylene glycol (PEG) and deoxyribonucleic acid sodium salt from salmon testes (ssDNA) were purchased from Sigma-Aldrich. DNA PCR Purification Kit was provided by Thermo-Fisher Scientific (Shanghai, China). Norcoclaurine, reticuline, scoulerine, cheilanthiofine, stylopine, protopine, and sanguinarine standards were purchased from Chengdu DeSiTe Biological Technology Co., Ltd (Chengdu, China). Phanta® Max Super-Fidelity DNA Polymerase and ClonExpress II One Step Cloning Kit were purchased from Vazyme (Nanjing, China). All restriction enzymes and T4 DNA ligase were purchased from NEB (Beijing, China).

### Strains and growth media
Yeast strains used in this study were listed in Supplementary Data 1. *S. cerevisiae* SAN006 was used as the parent strain for the production of sanguinarine[19]. *E. coli* DH5α (Tsingke Biotech, China) was used as the host to construct, maintain, and amplify plasmids. Recombinant *E. coli* strains were cultured in LB medium containing 100 mg L$^{-1}$ ampicillin or 50 μg/ml of kanamycin at 37 °C. Yeast strains were routinely cultivated in SED, containing 1.1 g L$^{-1}$ monosodium glutamate, 1.7 g L$^{-1}$ yeast nitrogen base without ammonium and amino acids, Complete Supplement Mixture (CSM; 1.2 g L$^{-1}$; MP Biomedicals) and 20 g L$^{-1}$ glucose, or YPD, containing 10 g L$^{-1}$ yeast extract, 20 g L$^{-1}$ peptone, and 20 g L$^{-1}$ glucose, at 30 °C. When necessary, G418 sulfate (Sangon Bio-tech Co., Ltd, Shanghai, China) was supplemented with a final concentration of 200 mg L$^{-1}$. All chemicals were from Sigma-Aldrich (St. Louis Missouri, USA), unless specifically mentioned.

### Plasmid construction
Cas9 expression plasmid (pRS41K-iCas9) and gRNA helper plasmids (pRS423-SpSgH and pRS426-SpSgH) were constructed in our previous studies[62,63]. Benchling CRISPR tool (https://benchling.com) was used to design gRNAs, which were cloned into *Bsa*I digested pRS423-SpSgH or pRS426-SpSgH. For the construction of multi-gRNA expression plasmids, individual gRNA expression cassettes were amplified by PCR and

subsequently assembled using Golden-Gate Assembly. The intein gene was cloned from the genome of BY4741. Site-directed mutagenesis of temperature-sensitive intein was performed by overlap extension PCR. Sanguinarine biosynthetic pathway genes (*EcBBE*, *CjBBE*, *McBBE*, *EcP6H*, *BUP1*, *XFPK*, *AROL*, and *BUP1*) were codon-optimized for yeast and synthesized by Genscript Biotech (Nanjing, China). The lycopene (*tHMG1*, *CrtE*, *CrtYB*, and *CrtI*) and sanguinarine biosynthetic pathway genes were cloned into the multiple cloning sites (MSCs) of the pESC series vectors (pESC-URA, pESC-LEU, pESC-TPR, and pESC-HIS) by digestion/ligation (MCS1: *Bam*HI/*Xho*I; MCS2: *Not*I) or Gibson Assembly. Oligonucleotide synthesis and DNA sequencing were performed by Tsingke Biotech (Hangzhou, China). Plasmids were extracted from *E. coli* using the AxyPrep Plasmid Miniprep Kit (Axygen, Shanghai, China). All the plasmids used in this study were listed in Supplementary Data 2. All the integration sites used in this study are listed in Supplementary Table 4. The coding sequences of the heterologous pathway genes were listed in Supplementary Data 3 and 4. The oligonucleotides used in this study were listed in Supplementary Data 5.

### Yeast transformation and strain construction
All genetic modifications in yeast were carried out via the CRISPR/Cas9-mediated genome editing method[64,65]. Heterologous gene expression cassettes were amplified by PCR with 40 bp homology arms to the target chromosomal loci and co-transformed with the corresponding gRNA plasmids to the Cas9 expressing yeast strains using the LiAc/ssDNA/PEG method[66]. Unless specifically mentioned, 600 ng gRNA plasmids and 600 ng of each donor DNA fragment were co-transformed.

### Growth curves
Growth curves were measured in triplicate in 24-well microtiter plates containing 3 mL of YP medium containing 2% glucose. Cultures were inoculated using saturated overnight cultures to an initial OD$_{600}$ of -0.2 and wrapped in Parafilm to minimize evaporation. Absorbance readings were taken at 600 nm every 4 h with a Tecan Infinite 200 PRO plate reader (Trading AG, Switzerland) over the course of 2 days.

### Fluorescence intensity measurement
A single colony of yeast strain was inoculated into 2 mL YPD medium and cultured until saturation (~48 h). The seed culture was then transferred to 2 mL fresh YPG medium with 1% inoculum and cultured for 30 h. Afterward, yeast cells were washed twice and resuspended in 10 mM PBS. Cell densities (OD$_{600}$) and mCherry fluorescence intensities (580/610 nm) were measured using a Tecan Infinite 200 PRO microplate reader (Trading AG, Switzerland). All yeast strains were cultured in 24 deep-well plates using a high-speed shaker (at 30 °C or 25 °C).

### Flow cytometry
Pre-cultured strains were transferred to 24 deep-well plates containing 2 mL of YPG medium with an initial OD$_{600}$ = 0.2. After 36 h of cultivation at 30 or 25 °C, cells were harvested by centrifugation at 3,542 g for 2 minutes and washed twice with PBS. The monoparametric detection of mCherry fluorescence in each sample was analyzed using a YL2 (561/620 nm) filter on an Attune NxT flow cytometer (Thermo Fisher Scientific). A total of 10,0000 cells were counted per sample to determine the percentage of fluorescent cells. GAL4-dINT cells were used to establish the negative boundary. In the subsequent experiments, the same boundary and gating strategy were applied to calculated fluorescent-positive cells.

### Transcriptome analysis
To conduct the transcriptome analysis, three single colony of SAN220-tsINT strains were cultured in SE medium with glycerol, galactose, and

a mixture of galactose and glycerol as the carbon source at 30 °C for 12 h, then switched the temperature to 25 °C and cultured for another 12 h. The total RNA of each sample was extracted by the TRIzol reagent (Invitrogen) according to the manufacturer's instructions. RNA with an integrity of more than 6.5 was detected by 2100 Bioanalyzer (Agilent Technologies) and adopted to perform library construction and sequencing. The complementary DNA libraries were constructed and sequenced via the Illumina NovaSeq6000 platform at the BioMarker Technologies Company (Beijing, China). Differential expression gene selection (DEGs) was performed using the DESeq2 R package (v.1.20.0) based on the count values of genes across different samples. The average values of three clones from each group were used, and a Fold Change ≥2 and FDR < 0.01 were employed as the selection criteria.

### Confocal microscopy
EGFP was fused to the *C*-terminus of the target proteins, whose encoding sequences were integrated into the wild-type yeast strain and verified by colony PCR. Strains were cultivated for 24 h and washed twice with 10 mM PBS. Then 2.5 μL of culture was applied to an objective glass, covered with a cover glass, and imaged immediately. Confocal images were captured using a Nikon Ti microscope with a 100× PlanAPO lens (NA 1.49), equipped with four diode lasers (405, 488, 555, and 639 nm). Samples were illuminated using high inclination laminated optical sheet TIRF illumination with 488 nm lasers, and its respective filter cube (Chroma). EGFP acquisition parameters were: 300–578 nm (emission wavelength range), 0.39 μs (line time), 600–1000 (gain), and 0 (offset). A line average of eight was applied to both channels. Images were processed with NIS-Elements 2.

### Fed-batch fermentation in bioreactors
After overnight cultivation in YPD medium at 30 °C, seed cultures were inoculated (10%, v/v) into 1.2 L vessels using the T&J-Minibox5 Intelli-Ferm Parallel Bioreactors System (T&J Bio-engineering Co. Ltd., Shanghai, China) with 0.6 L working volume. The fermentation medium comprised 5.1 g L$^{-1}$ of yeast nitrogen base without ammonium and amino acids, 5 g L$^{-1}$ of monosodium glutamate, and 10 g L$^{-1}$ of glycerol. The agitation speed was adjusted with a constant air input flow rate of 3.0 vvm, to maintain the dissolved oxygen at 20% of air saturation. 15% (w/v) Acetic acid and 15% (w/v) ammonium hydroxide were automatically added to maintain pH at 5.0. After 59 h, the temperature was switched from 30 °C to 25 °C to initiate the expression of heterologous genes. 400 g L$^{-1}$ galactose and 60 g L$^{-1}$ yeast extract were separately supplemented as feed while maintaining the ethanol concentration in the fermentation broth below 5 g L$^{-1}$.

### Qualitative and quantitative analysis of BIAs
Metabolites were extracted from a culture broth containing cells and a growth medium. 100 μL culture broth was combined with 900 μL 100% acetonitrile (ACN), which was subjected to high-speed shaking to disrupt yeast cells, followed by centrifugation at 14,167 g for 10 minutes to fully separate the supernatant. To avoid solvent effects, 50 μL supernatant was subsequently transferred to 400 μL water, leading to a final concentration of 10% ACN. This method corresponds to a 90-fold dilution of the sample, and additional dilution multiples were necessary for more productive BIA strains and fed-batch fermentation broth.

For the separation and analysis of the treated samples, 2 μL was injected into a 1290 Infinity II LC system (Agilent Technologies) equipped with a Zorbax Rapid Resolution HT C18 column (30 × 2.1 mm, 1.8 μm; Agilent Technologies). Metabolites were separated using the following gradient: from 5% B to 85% B from 0 to 1 min (0.2 mL min$^{-1}$), held at 85% B from 1 to 3.5 min (0.2 mL min$^{-1}$), 85% B to 5% B from 3.5 to 4.5 min (0.2 mL min$^{-1}$), and held at 5% B from 4.5 to 5 min (0.2 mL min$^{-1}$). A post-time of 2 min was employed. Solvent A was 0.1% formic acid in water, and solvent B was 0.1% formic acid in 100% ACN. The eluent from the LC system was injected into a 6470 triple quadrupole LC/MS system (Agilent Technologies) equipped with an ESI ion source. The resolution, capillary voltage, and source temperature were set to 100,000, 4 kV, and 350 °C, respectively. The UPLC-MS data were processed and analyzed using MassHunter software (Agilent Technologies). LC-MS multiple reaction monitoring (MRM) mode was employed for the quantitative analysis of sanguinarine and its intermediates and detailed parameters are listed in Supplementary Table 5. The 6470 triple quadrupole LC/MS system in production mode and the 6545 high-resolution mass spectrometry LC/Q-TOF in Targeted MS/MS mode detection were used for qualitative analysis of halogenated derivatives, with detailed parameters listed in Supplementary Table 2, 3, and 6.

### Statistics and reproducibility
All experimental data were at least in triplicate and expressed as mean ± standard error. All data analyses were performed by Excel or GraphPad Prism 8.

### Reporting summary
Further information on research design is available in the Nature Portfolio Reporting Summary linked to this article.

## Data availability
The transcriptome sequencing data generated in this study have been deposited into NCBI with an accession number of PRJNA1108474. All the protein structures are obtained from AlphaFold 2.0. Source data are provided in this paper.

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

## Acknowledgements

This work was supported by the National Key Research and Development Program of China (2018YFA0901800 to J.L., 2021YFC2103200 to J.L., and 2023YFC3402400 to L.H.), the National Natural Science Foundation of China (22278361 to J.L., 32200052 to C.D., and 32300053 to F.X.), China Postdoctoral Science Foundation (2023M733096 to J.G.), Fundamental Research Funds for the Zhejiang Provincial Universities (226-2023-00015 to J.L.), and Fundamental Research Funds for the Central Universities (226-2022-00214 to J.L.). We would like to thank Prof. Hongwei Yu and Lidan Ye from Zhejiang University for kindly sharing GAL4M9. We also would like to thank iBioFoundry and Core Facility at ZJU-Hangzhou Global Scientific and Technological Innovation Center for analytical support.

## Author contributions

J.L. conceptualized and supervised the study. Y.G. performed the experiments, analyzed data, and drafted the manuscript. Z.B., L.H., Z.X., and J.L. revised the manuscript. M.Z. contributed to global metabolic optimization at iBioFoundry. Y.G. and C.Y. performed fluorescence microscopy. Y.G., J.Z., Y.Z., and J.G. conducted the validation of SIM-TeGES for multiple proteins in different hosts. Y.G. and C.S. performed western blot analysis. Y.G., G.L., and H.D. performed molecular dynamics simulations and analyzed the mechanism of intein splicing. Y.G., M.L., and F.X. performed the bioreactor fermentation. Y.G. and D.L. performed the establishment of qualitative and quantitative analytical methods for BIAs. Y.G. and C.D. performed flow cytometry. Y.G. and D.G. performed qPCR analysis. All authors approved the manuscript.

## Competing interests

The authors declare no competing interests.
