## [Peer Review File · Nature Communications]

Intein-mediated Temperature Control for Complete Biosynthesis of Sanguinarine and its Halogenated Derivatives in YeastReviewers' Comments:

Reviewer #1:

Remarks to the Author:

In this study, Gou and colleagues aimed to improve the production of sanguinarine in baker's yeast, building on their previous research. To uncouple cell growth from product biosynthesis, the researchers devised a splicing intein-mediated temperature-responsive gene expression system (SIMTeGES) to regulate gene expression. An intein sequence was inserted into the coding sequence of GAL4 to create a temperature-sensitive transcription factor. The regularity of this engineered GAL4-tsINT was assessed through lycopene biosynthesis and fluorescent protein expression, and its mode of action was analyzed using *in silico* methods. Next, two BBE-like proteins, CjBBE and McBBE, were identified through database mining and engineered by replacing their N-terminal peptide with maltose-binding protein. This led to an improvement in the conversion of reticuline to scoulerine compared to the wild-type PsBBE. Furthermore, the team analyzed the protopine 6-hydroxylases and engineered its transmembrane domains by creating a chimeric enzyme EcCFS1-83-McP6Hs 84-522. This resulted in enhanced protein expression and improved localization. Finally, the team optimized the precursor supply and made other routine improvements. All of these efforts led to the achievement of the highest reported level of sanguinarine production. With this foundation, the authors were able to produce fluorinated sanguinarines by introducing fluorinated precursors into the cell chassis. Overall, this work demonstrates how to improve the production of plant natural products by comprehensively utilizing pathway engineering and metabolic engineering. The workload for this project is substantial, and the outcomes are good. Publication in Nature Communications is recommended, but the following concerns need to be addressed.

1. The current manuscript is lengthy, and it would be better to reduce the text by 10-20% and eliminate redundancy.
2. It appears that downstream enzymes cannot tolerate the halogen substitution, and only fluorine can be transferred to the end product. The authors should briefly discuss it. In addition, the characterization of F-sanguinarin is based solely on MS analysis. It is necessary to verify the chemical structure using NMR.
3. It should cite the Smokle's works on noscapine and halogenated alkaloids (PNAS doi: 10.1073/pnas.1721469115)

Reviewer #2:

Remarks to the Author:

The authors report the strategy of the engineered yeast to produce sanguinarine, a potential antimicrobial and anti-cancer compound, achieving a high yield of 715.12 mg/L. A temperature-responsive gene expression system was developed to mitigate cytotoxic effects and enhance production. The work paves the way for scalable yeast-based production of various benzyloisoquinoline alkaloids and their derivatives.

It is supported by detailed research and the text is well theorised and described. The following points could be improved.

- 1) In fed-batch fermentation, sanguinarine production increased 7-fold to 700 mg/L. Is this just an increase in biomass compared to flask fermentation? If not, you should discuss this and show data comparing the biomass in flask fermentation to prove that it is not just an increase in biomass. Also, is it possible to show the validity of SIMTeGES by determining the production of sanguinarine under normal culture conditions in fed-batch fermentation?
- 2) Reference 31 produced 4.6 g/L of reticuline. If possible, the effectiveness of SIMTeGES should be demonstrated by determining the production of reticuline in the present system. Alternatively, is SIMTeGES not effective in producing reticuline without the toxic effects of the compounds produced?

3) In Figures 3e and 3f, even at non-permissive temperatures, spliced 35 kDa bands are observed along with unspliced 85 kDa bands. And the ratio of spliced to unspliced forms does not appear to match the fluorescence in Figure 3d. Therefore, in Figures 3e and 3f, the quantification of the ratio of spliced to unspliced forms are necessary, and then the correlation between the ratio and the value of fluorescence should be discussed.

4) In Figures 6c and 7c, the images are too small to be easily seen, so it is better to include an enlarged image of the cells as insets.

5) In figures, the numbers a-d should be written in upper or lower case. (They are capitalised in the text).

6) In Fig. 1, an explanation of the abbreviations NOR, RET, SCO, CHE, PRO and SAN is given in the diagram, but also in the legend of Fig. 4. If this is to be adapted, the explanation of the abbreviations should also be given in the legend of Figs. 5 and 6.

7) In Fig. 2D, SED, SEDG, SEG, SEGly and SEGlyG need to be explained.

8) Page 18, line 385: Fig. 5C should be corrected to Fig. 5B.

Page 18, line 391: Fig. 5D should be corrected to Fig. 5C.

9) Page 23, line 487: Is Fig. 7B a C? Please check the citations in Fig. 7A-E as the text and figures do not match.

Reviewer #3:

Remarks to the Author:

Gou et al show a novel Splicing Intein-Mediated Temperature-responsive Gene Expression System, which they call SIMTeGES that enables regulation of expression of heterologous genes. Their system uses temperature as a regulatory signal to separate cell growth from product biosynthesis processes (e.g. for a two-stage process). At 30 °C, optimal for yeast growth, SIMTeGES keeps the GAL4 transcription factor inactive, which allows cell growth. Conversely, at 25 °C, GAL4 is activated by splicing and this promotes the expression heterologous genes under control of the GAL4 responsive promoter. The approach was used to separate cell growth and sanguinarine biosynthesis (a toxic compound for yeast). Additionally the authors show that its generally applicable and facilitates the temperature-controlled expression of additional target proteins (e.g., mCherry, GAL80).

The case study of sanguinarine biosynthesis is impressive as it involves heterologous expression of many enzymes and optimization of several steps. The authors report the highest titer achieved for sanguinarine. The topic of two-stage process and dynamic control of microbial systems is very relevant and the manuscript is well written. I have few points that should be considered:

1) The manuscript combines two important developments: the SIMTeGES, and the sanguinarine production. The results are structured in this way, but Figure 1 starts with the sanguinarine pathway. I recommend to present the SIMTeGES first. In Figure 1, the expression system should be explained: which genes are under which promoter, where are they integrated into the genome. This would help to follow the metabolic engineering strategy that is described in the second part of the result section.

2) The authors should describe how the temperature sensitive (TS) intein VMAL212P was derived. Did they screen for the TS phenotype, did they find additional TS alleles?

3) It seems that the GAL 4M9 and GAL4 tsINT perform similar for lycopene and mCherry, and the largest benefit is visible in the sanguinarine case. The authors should better compare the pros and cons of these system also in terms of repression efficiency at 30°C. The Western Blot indicates significant splicing at 30°C for GAL4 tsINT.

4) In Figure 6a the authors show a very nice example of metabolic engineering (or bottleneck

identification) based on metabolite accumulation. However, the presentation is very basic. The authors should provide quantitative data (e.g. fold-changes) for the metabolites together with abundances of the respective enzymes. This would help to understand the systematic approach for changes in the pathway.

5) The authors use complex medium for their bioprocess for sanguinarine biosynthesis. At the time point of sanguinarine synthesis the main carbon source glycerol and galactose are depleted. Therefore, the authors should show which carbon source goes into sanguinarine biosynthesis. It is not clear if tyrosine fed to the cultures in Figure 6 or if tyrosine was derived from the complex medium.

Reviewer #1 (Comments for the Author):

In this study, Gou and colleagues aimed to improve the production of sanguinarine in baker's yeast, building on their previous research. To uncouple cell growth from product biosynthesis, the researchers devised a splicing intein-mediated temperature-responsive gene expression system (SIMTeGES) to regulate gene expression. An intein sequence was inserted into the coding sequence of GAL4 to create a temperature-sensitive transcription factor. The regularity of this engineered GAL4-tsINT was assessed through lycopene biosynthesis and fluorescent protein expression, and its mode of action was analyzed using *in silico* methods. Next, two BBE-like proteins, CjBBE and McBBE, were identified through database mining and engineered by replacing their N-terminal peptide with maltose-binding protein. This led to an improvement in the conversion of reticuline to scoulerine compared to the wild-type PsBBE. Furthermore, the team analyzed the protopine 6-hydroxylases and engineered its transmembrane domains by creating a chimeric enzyme EcCFS1-83-McP6Hs 84-522. This resulted in enhanced protein expression and improved localization. Finally, the team optimized the precursor supply and made other routine improvements. All of these efforts led to the achievement of the highest reported level of sanguinarine production. With this foundation, the authors were able to produce fluorinated sanguinarines by introducing fluorinated precursors into the cell chassis. Overall, this work demonstrates how to improve the production of plant natural products by comprehensively utilizing pathway engineering and metabolic engineering. The workload for this project is substantial, and the outcomes are good. Publication in Nature Communications is recommended, but the following concerns need to be addressed.

Thanks for the reviewer's positive comments on our manuscript.

1. The current manuscript is lengthy, and it would be better to reduce the text by 10-20% and eliminate redundancy.

Thanks for the reviewer's suggestion. We agree with the reviewer that our manuscript is lengthy, as we tried our best to include as much info as possible. Accordingly, we have taken steps to address this issue, such as revising the Introduction and Discussion sections, moving some materials and methods into the supporting information, and trimming redundant background information at the beginning of each result section. Additionally, we've deleted some redundant results that didn't contribute significantly to the completeness of the present work. These adjustments are in the Lines 54-55, 204-209, 380-382, 424-427, 458-462, 538-423, 559-615, 622-633, 653-664 of the original manuscript and are intended to enhance the readability and clarity of our manuscript for readers.

2. It appears that downstream enzymes cannot tolerate the halogen substitution, and only fluorine can be transferred to the end product. The authors should briefly discuss it. In addition, the characterization of F-sanguinarin is based solely on MS analysis. It is necessary to verify the chemical structure using NMR.

Thanks for the reviewer's comments. We agree with the reviewer that it is better to briefly discuss the characteristics of halogenated derivatives entering the sanguinarine pathway. Enzyme promiscuity confers adaptability to the yeast cell factory, allowing it to accommodate substrates like halogenated tyrosine, aiding exploration of new chemical spaces for drug discovery. However, halogenated derivatives, not natural substrates of the pathway enzymes, exhibit lower conversion yield, especially in longer pathways. Our current results leave some ambiguity regarding whether downstream enzymes, such as reticuline oxidase (BBE), cannot tolerate chlorine substitution or there's an insufficient supply of the precursor Cl-reticuline, resulting in undetectable Cl-scoulerine. Fluorination, owing to its reduced steric hindrance, appears to be more readily integrated into the pathway compared to chlorination and iodination. Similar observations have been noted in the incorporation of carbon-halogen bonds into alstonine (5 steps for halogenated alstonine biosynthesis from halogenated tryptophan), berberine (9 steps for F-tetrahydrocolumbamine biosynthesis from 3-fluorotyrosine), and noscapine (7 steps for halogenated reticuline biosynthesis from halogenated tyrosines) pathways (Bradley, S.A. et al. *Nat Chem Biol* 19, 1551–1560 (2023); Han, J., Li, S. *Commun Chem* 6, 27 (2023); Li, Y. et al *Proc Natl Acad Sci U S A* 115, 17 (2018)). Leveraging the efficient biotransformation rate of yeast cellular factories, we successfully detected F-sanguinarine, with the most biocatalytic steps (15 steps) for generating fluorinated derivatives. Therefore, we have incorporated additional viewpoints in Lines 553-562 of the revised manuscript.

Regarding the structural characterization of F-sanguinarine, as suggested by the reviewer, we agree with the reviewer to recognize NMR as the most reliable approach. However, the low yield of the new-to-nature compound F-sanguinarine, currently only detectable, poses challenges in isolating and purifying sufficient quantities for NMR analysis. Similar studies relied solely on mass spectrometry for accurate molecular weight analysis as well, such as Bradley, S.A. et al. *Nat Chem Biol* 19, 1551–1560 (2023), Han, J., Li, S. *Commun Chem* 6, 27 (2023), and Li, Y. et al *Proc Natl Acad Sci U S A* 115, 17 (2018).

Nevertheless, we have devoted every effort to systematically characterize F-sanguinarine. Initially, we compared the MS² spectra of sanguinarine and F-sanguinarine using triple quadrupole (QQQ) mass spectrometry (Fig. 7c), confirming

the same molecular weight differences in precursor ($[F]^- = 18$, sanguinarine $[M]^+ = 332$, F-sanguinarine $[M]^+ = 350$) and characteristic fragments (sanguinarine: 245.9, 273.9; F-sanguinarine: 263.9, 291.9). Building on this, we further analyzed F-sanguinarine's exact molecular weight (accurate to 4 decimal places, Fig. 7e and Supplementary Fig. S41) and isotopic distribution (Supplementary Table S3) using high resolution qTOF mass spectrometry. By comparing with the sanguinarine standard, analyzing the exact molecular weight, and considering isotopic distributions, we believe our conclusion is well-supported. Accordingly, we have revised the manuscript to provide a more systematic description of characterizing F-sanguinarine in Lines 493-495 of the revised manuscript and added the isotopic information of precursor and characteristic fragments for F-sanguinarine (Supplementary Table S3). Qualitative results analyzed using MassHunter software (Agilent Technologies) are summarized in the source data.

Supplementary Table S3 The isotopic distribution of F-sanguinarine

m/z (observed)	Species	Predicted formula	Isotope
350.0838	Precursor A, $[M]^+$	$C_{20}H_{13}FNO_4$	351.0816
321.0724	Fragment B, $[M]^+$	$C_{19}H_{12}FNO_3$	322.0756, 323.0881, 324.0878
291.0620	Fragment C, $[M]^+$	$C_{18}H_{10}FNO_2$	292.0659, 293.0751, 294.0868
263.0676	Fragment D, $[M]^+$	$C_{17}H_{10}FNO$	264.0724, 265.0808
249.0517	Fragment E, $[M]^+$	$C_{16}H_8FNO$	250.0549, 251.0631
233.0569	Fragment F, $[M]^+$	$C_{16}H_8FN$	234.0607, 235.0203

3. It should cite the Smokle's works on noscapine and halogenated alkaloids (PNAS doi: 10.1073/pnas.1721469115)

Thanks for the reviewer's suggestion. Smokle's works on noscapine and halogenated alkaloids indeed represent the first examples of feeding halogenated tyrosine for the biosynthesis of halogenated BIAs. We have incorporated the referenced publication in the introduction and discussion sections of the revised manuscript, appearing in Lines 94, 103, and 557, respectively.

Reviewer #2 (Comments for the Author):

The authors report the strategy of the engineered yeast to produce sanguinarine, a potential antimicrobial and anti-cancer compound, achieving a high yield of 715.12 mg/L. A temperature-responsive gene expression system was developed to mitigate cytotoxic effects and enhance production. The work paves the way for scalable yeast-

based production of various benzyloisoquinoline alkaloids and their derivatives.

It is supported by detailed research and the text is well theorised and described. The following points could be improved.

Thanks for the reviewer's positive comments on our manuscript.

1. In fed-batch fermentation, sanguinarine production increased 7-fold to 700 mg/L. Is this just an increase in biomass compared to flask fermentation? If not, you should discuss this and show data comparing the biomass in flask fermentation to prove that it is not just an increase in biomass.

Also, is it possible to show the validity of SIMTeGES by determining the production of sanguinarine under normal culture conditions in fed-batch fermentation?

Thanks for the reviewer's comments. As depicted in Supplementary Fig. S26, the OD₆₀₀ of the SAN231 strain in shake flasks was ~10, while in the fermenter, the average OD₆₀₀ reached ~130, representing a substantial increase in biomass by ~13-fold during fed-batch fermentation. This significant rise in biomass indeed contributes to the overall improvement in sanguinarine production (~7-fold). Nevertheless, we would like to revise the fed-batch fermentation results. Despite repeated attempts post-submission, only in a few cases could we achieve the results with the sanguinarine titer of ~700 mg L⁻¹ and OD₆₀₀ of ~130. Most attempts resulted in the sanguinarine titer of ~450 mg L⁻¹ and OD₆₀₀ of ~80. Therefore, we have updated Figures 6b and 6c with these more representative results. We acknowledge that this discrepancy could potentially stem from additional limitations in sanguinarine production, possibly due to cytotoxicity effects. Anyway, in the case of new results, biomass was increased ~8-fold, while sanguinarine titer was increased by ~4.4-fold. In other words, sanguinarine yield per cell during fed-batch fermentation is lower compared with that in shake flasks, indicating that sanguinarine production was still limited by its cytotoxicity and fermentation conditions should be further optimized. To address these challenges, we are incorporating relevant strategies such as transporter engineering and tolerance engineering into our ongoing research efforts.

To further demonstrate the validity of SIMTeGES by determining sanguinarine production under normal culture conditions, we replaced GAL4-tsINT in strain SAN231 with the wild-type GAL4. To ensure the stability of the strain due to leakage expression of the GAL promoter under low glucose concentrations, we also complemented the transcriptional repressor GAL80, resulting in the construction of strain SAN334. Subsequently, we assessed the sanguinarine production capability of strain SAN334 in shake flasks at both 25 °C and 30 °C, which was compared with strain

SAN231. The results indicated minimal variation in sanguinarine production for SAN334 strain at different temperatures, with titers averaging around 26 mg L⁻¹, representing a decrease of approximately 4-fold when compared with strain SAN231 (Supplementary Fig. S26). This observation underscores the significance of SIMTeGES in enhancing sanguinarine biosynthesis, as highlighted in Lines 446-449 of the revised manuscript. Given the significantly lower sanguinarine production using strain SAN334, we think it unnecessary to perform fed-batch fermentation under normal culture conditions.

Supplementary Fig. S26 Validation of SIMTeGES by replacing GAL4-tsINT of strain SAN231 with the wild-type GAL4. Strain SAN334 was cultured in shake flasks at both 25 °C and 30 °C. Abbreviations not previously defined: NOR, (*S*)-Norcoclaurine; RET, (*S*)-Reticuline; SCO, (*S*)-Scoulerine; CHE, (*S*)-Cheilanthifoline; PRO, Protopine; SAN, Sanguinarine. Data are presented as mean ± s.e.m. (n = 3 biologically independent samples). Statistical analysis was performed using two-way ANOVA followed by Tukey's multiple comparisons test (ns, not significant, **p* < 0.033, ***p* < 0.002, ****p* < 0.001).

2. Reference 31 produced 4.6 g/L of reticuline. If possible, the effectiveness of SIMTeGES should be demonstrated by determining the production of reticuline in the present system. Alternatively, is SIMTeGES not effective in producing reticuline without the toxic effects of the compounds produced?

Thanks for the reviewer's comments. We agree with the reviewer that the effectiveness of SIMTeGES should be further demonstrated in producing compounds without toxic

effects. Although our study has already showcased its significant efficacy and adaptability in facilitating non-toxic lycopene biosynthesis, with GAL4-tsINT exhibiting a 15.5-fold increase in lycopene production over GAL4M9 and a 3.7-fold increase over wild-type GAL4 (Supplementary Fig. S7).

Regarding reference 31 highlighted by the reviewer, where the highest titer of reticuline (4.6 g L⁻¹) was reported using pulsed fed-batch fermentation, direct comparison of reticuline production becomes complex due to differences in experimental parameters such as promoters (constitutive, e.g., *P_{TDH3}*, *P_{TEF1}*, *P_{PGK1}*), culture media (2×SC), and carbon sources (sucrose). Additionally, the specialized pulsed fed-batch fermentation method employed by the authors resulted in a 10.29-fold increase in titer from 447 mg L⁻¹ to 4.6 g L⁻¹. Nevertheless, to address the reviewer's concern, we evaluated reticuline accumulation under different carbon sources based on the SAN220, SAN220-M9, and SAN220-tsINT strains with the *PsBBE* gene knocked out. As illustrated in the figure below, when both glycerol and galactose were present (SEGlyG), GAL4-tsINT exhibited the highest reticuline titer of 270.15 mg L⁻¹, surpassing GAL4 by 1.89-fold and GAL4M9 by 2.00-fold, respectively (Fig. R1).

Fig. R1 Evaluation of the accumulation of intermediate reticuline using SIMTeGES-GAL4 under diverse carbon source conditions. SED, SEDG, SEG, SEGly, and SEGlyG denote different fermentation conditions in SE medium, with the following carbon sources: 20 g L⁻¹ glucose, a combination of 20 g L⁻¹ glucose and 10 g L⁻¹ galactose, 20 g L⁻¹ galactose, 20 g L⁻¹ glycerol, and a combination of 10 g L⁻¹ glycerol and 20 g L⁻¹ galactose, respectively. Data are presented as mean ± s.e.m. (n = 3 biologically independent samples). Statistical analysis was performed using two-way ANOVA followed by Tukey's multiple comparisons test (ns, not significant, **p* < 0.033, ***p* < 0.002, ****p* < 0.001).

Given our focus on optimizing sanguinarine production and the occasional need for controlled accumulation levels of reticuline as an intermediate metabolite, achieving equivalent or superior reticuline yields poses challenges without a similar fermentation system (pulsed fed-batch fermentation). However, considering the performance across multiple systems, we speculate that SIMTeGES holds promise for enhancing the biosynthesis of diverse target products by decoupling cell growth from product synthesis, thereby alleviating cellular metabolic burdens and fostering efficient target product biosynthesis.

3. In Figures 3e and 3f, even at non-permissive temperatures, spliced 35 kDa bands are observed along with unspliced 85 kDa bands. And the ratio of spliced to unspliced forms does not appear to match the fluorescence in Figure 3d. Therefore, in Figures 3e and 3f, the quantification of the ratio of spliced to unspliced forms are necessary, and then the correlation between the ratio and the value of fluorescence should be discussed. Thanks for the reviewer's comments. We agree with the reviewer that quantifying the ratio of spliced to unspliced forms of the protein in Figures 3e and 3f is important, although our primary aim was to employ western-blot to validate the direct correlation between protein activity and intein splicing. We conducted grayscale analysis of the bands corresponding to mCherry-tsINT at 35 kDa (spliced form) and 85 kDa (unspliced form) under permissive or non-permissive conditions using Image J software. The ratio of unspliced to spliced protein under permissive and non-permissive conditions is 1:5.32 and 4.59:1, respectively (Fig. R2). As noted by the reviewer, there is indeed a higher abundance of spliced protein under non-permissive conditions compared to the fluorescence results. It is noteworthy that the reduction in the 85 kDa protein under permissive conditions is much lower than the increase in the 35 kDa protein. We speculate two main reasons for the observed discrepancy: firstly, intein splicing is a cofactor or energy-independent intramolecular process mainly through bond rearrangement. Although we expedited the sample preparation process as quickly as possible, it still takes longer than fluorescence detection (flow cytometry and fluorescence microplate reader), and protein samples under non-permissive conditions are likely to undergo partial splicing during sample preparation (room temperature is 25 °C). Secondly, the transmembrane efficiency of large molecular weight proteins is relatively low, and there are differences in grayscale values corresponding to proteins of different molecular weights. Therefore, western-blot results can only serve as semi-quantitative references and are more suitable for proving the conclusion that temperature-dependent activity relies on intein splicing. Accordingly, to prevent potential misunderstanding, we choose not to put the grayscale analysis results in the manuscript.

To more accurately characterize the activity and leakage level of mCherry-tsINT, we have supplemented the corresponding flow cytometry results (Supplementary Fig. S11). The leakage expression level of mCherry-tsINT under non-permissive conditions is only 5.21%, while the positivity rate under permissive conditions is 99.8%, further demonstrating that SIMTeGES exhibits strong activity and low leakage expression levels. We have incorporated these results into Lines 245-248 of the revised manuscript.

Fig. R2: Grayscale analysis results of mCherry-tsINT at 35 kDa (spliced form) and 85 kDa (unspliced form) under permissive or non-permissive conditions in Figures 3e and 3f, respectively.

Supplementary Fig. S11 Flow cytometry to further investigate the activity and leaky expression of mCherry-tsINT.

4. In Figures 6c and 7c, the images are too small to be easily seen, so it is better to include an enlarged image of the cells as insets.

Thanks for the reviewer's comments. I think the reviewer tends to mean Fig. 4d and 5c, where confocal microscope images are provided. We agree with the reviewer that the cell images are too small for easy viewing. To improve the clarity of cell localization and fluorescence, we have enlarged the cell images in Fig. 4d and 5c. In addition, for the reviewer's reference, we keep the original full-scale images in the revised Source Data file.

5. In figures, the numbers a-d should be written in upper or lower case. (They are capitalised in the text).

Thanks for the reviewer's comments. To ensure consistency with the figure numbering in the text, we have changed all figure labels in the manuscript to lower case.

6. In Fig. 1, an explanation of the abbreviations NOR, RET, SCO, CHE, PRO and SAN is given in the diagram, but also in the legend of Fig. 4. If this is to be adapted, the explanation of the abbreviations should also be given in the legend of Figs. 5 and 6.

Thanks for the reviewer's suggestion. We have already provided the explanation of the abbreviations in the legends of Fig. 6, as indicated in Lines 999-1002 of the revised manuscript. Additionally, we have updated the abbreviations (PRO and SAN) in Fig. 5 of the revised manuscript to their full names.

7. In Fig. 2D, SED, SEDG, SEG, SEGly and SEGlyG need to be explained.

Thanks for the reviewer's comments. We have included explanations for the abbreviations SED, SEDG, SEG, SEGly, and SEGlyG in the legend of Fig. 1d of the revised manuscript (Fig. 2d of the original manuscript), as outlined in Lines 930-933 of the revised manuscript. Additionally, we have provided explanations for the abbreviations YPD, YPDG, YPG, YPGly, and YPGlyG in Supplementary Fig. S7.

8. Page 18, line 385: Fig. 5C should be corrected to Fig. 5B.

Page 18, line 391: Fig. 5D should be corrected to Fig. 5C.

Thanks for the reviewer's comments. We have made the necessary corrections to the figure labels in the revised manuscript, changing "Fig. 5c" to "Fig. 5b" in Page 18, Line 384, and "Fig. 5d" to "Fig. 5c" in Page 18, Line 390 of the revised manuscript.

9. Page 23, line 487: Is Fig. 7B a C? Please check the citations in Fig. 7A-E as the text and figures do not match.

Thanks for the reviewer's comments. We agree with the reviewer that the citations in

Fig. 7a-e not aligning with the figures. We have corrected this by updating the citation in Page 23, Line 488 from "Fig. 7b" to "Fig. 7c and 7e" and have meticulously reviewed the citation issues concerning Fig. 7a-e in the revised manuscript.

Reviewer #3 (Comments for the Author):

Gou et al show a novel Splicing Intein-Mediated Temperature-responsive Gene Expression System, which they call SIMTeGES that enables regulation of expression of heterologous genes. Their system uses temperature as a regulatory signal to separate cell growth from product biosynthesis processes (e.g. for a two-stage process). At 30 °C, optimal for yeast growth, SIMTeGES keeps the GAL4 transcription factor inactive, which allows cell growth. Conversely, at 25 °C, GAL4 is activated by splicing and this promotes the expression heterologous genes under control of the GAL4 responsive promoter. The approach was used to separate cell growth and sanguinarine biosynthesis (a toxic compound for yeast). Additionally the authors show that its generally applicable and facilitates the temperature-controlled expression of additional target proteins (e.g., mCherry, GAL80).

The case study of sanguinarine biosynthesis is impressive as it involves heterologous expression of many enzymes and optimization of several steps. The authors report the highest titer achieved for sanguinarine. The topic of two-stage process and dynamic control of microbial systems is very relevant and the manuscript is well written. I have few points that should be considered:

Thanks for the reviewer's positive comments on our manuscript.

1. The manuscript combines two important developments: the SIMTeGES, and the sanguinarine production. The results are structured in this way, but Figure 1 starts with the sanguinarine pathway. I recommend to present the SIMTeGES first. In Figure 1, the expression system should be explained: which genes are under which promoter, where are they integrated into the genome. This would help to follow the metabolic engineering strategy that is described in the second part of the result section.

Thanks for the reviewer's comments. We agree with the reviewer that presenting the SIMTeGES first is beneficial. Consequently, we have reorganized the manuscript to feature SIMTeGES first, with the sanguinarine pathway now described in Fig. 3 of the revised manuscript. We believe this adjustment will offer a clearer delineation of our two main developments.

Due to space limitations, we don't think it possible to include so much detailed info

(integration loci, promoters, and genes) in Fig. 3 of the revised manuscript (Fig. 1 of the original manuscript). Instead, we have included the construction processes of the SIMTeGES and sanguinarine bioproduction-related strains in Supplementary Fig. S1 and S14 to further explain the expression system. These figures detail which genes are regulated by which promoter and where they are integrated into the yeast genome. Additionally, in conjunction with the yeast strains and cell information in Table 1 and plasmid information in Supplementary Table S6, we have also provided detailed information about the integration sites in Supplementary Table S7. These additions hopefully enhance the understanding of the metabolic engineering strategy described in the results section.

Supplementary Fig. S1 Construction procedures for SIMTeGES-related strains. Detailed depiction of the integrated gene cassette with the utilized promoters and their corresponding integration sites. Detailed information regarding the integration sites can be found in Supplementary Table S7.

Supplementary Fig. S14 Construction procedures for sanguinrine bioproduction-related strains. Detailed depiction of the integrated gene expression cassettes with the promoters and their corresponding integration sites. Additional information regarding the integration sites can be found in Supplementary Table S7.

2. The authors should describe how the temperature sensitive (TS) intein VMAL212P was derived. Did they screen for the TS phenotype, did they find additional TS alleles? Thanks for the reviewer's comments. We agree with the reviewer that providing further clarification on the derivation of the temperature-sensitive intein VMAL212P is necessary. Indeed, we valuated several intein variants (Zeidler, M., et al. Nat Biotechnol 22, 871–876 (2004). doi: 10.1038/nbt979; Tan G., et al. Genetics 183, 13–22 (2009). doi: 10.1534/genetics.109.104794) and found that only the L212P mutant exhibited a distinct temperature-sensitive phenotype under a modest temperature difference (5 °C, Supplementary Fig. S4). While other mutants have been reported to show different temperature-sensitive characteristics, they did not meet the specific requirements for our application. Such discrepancy may result from the use of different expression system: plasmid expression in previous studies and genome integrated expression in the present study. Consequently, we focused our investigation primarily on the VMA^{L212P} variant (a.k.a. tsINT), applying it to different target proteins (mCherry, GAL80, and GAL4) and in various host systems (*S. cerevisiae*, *P. pastoris*, and mammalian cells). Additionally, we utilized molecular dynamics (MD) simulations to gain insights into the temperature dependence mechanism. To address the reviewer's concern, we have incorporated additional details regarding the intein mutant screening process in Lines 157-159 of the revised manuscript.

Supplementary Fig. S4 Characterization of temperature-sensitive intein variants. (a) Sequence features of intein and intein variants. (b) The colored metabolite lycopene serves as a reporter system, reflecting GAL4 activity for screening appropriate intein variants. Three single clones for each strain were simultaneously spotted onto two YPG plates and incubated at 25 °C and 30 °C for 2 days, respectively.

3. It seems that the GAL 4M9 and GAL4 tsINT perform similar for lycopene and mCherry, and the largest benefit is visible in the sanguinarine case. The authors should better compare the pros and cons of these system also in terms of repression efficiency at 30° C. The Western Blot indicates significant splicing at 30°C for GAL4 tsINT.

Thanks for the reviewer's comments. We agree with the reviewer that a more comprehensive comparison of the pros and cons of the two temperature-responsive systems would enhance the understanding of our developed SIMTeGES. In summary, SIMTeGES was employed to achieve temperature-controlled expression of *GAL4*, *GAL80*, and *mCherry*. Specifically, in the case of SIMTeGES-GAL4, we utilized GAL4-tsINT and the previously reported temperature-sensitive mutant GAL4M9 to regulate the expression of the *mCherry* reporter gene, as well as the lycopene and sanguinarine pathway, enabling a thorough evaluation of their activities, leakage expression levels, and temperature sensitivities.

Upon the use of these temperature control systems to regulate *mCherry* expression, we validated their temperature-sensitive characteristics at both 25 °C and 30 °C. At 25 °C, both systems effectively initiated *mCherry* expression. The fluorescence intensity of GAL4-tsINT was 1.28 times higher than that of GAL4M9, while at 30 °C, the leakage level of GAL4-tsINT was slightly higher than that of GAL4M9, measuring 5.74% and 1.79% respectively (Fig. 1c and Supplementary Fig. S5 of the revised manuscript). When assessing temperature-responsive kinetics (involving the transitioning from 30 °C to 25 °C), GAL4-tsINT consistently exhibited stronger fluorescence changes, reaching 26.2-fold, whereas GAL4M9 only changed 15.4-fold (Supplementary Fig. S6). In fact, in the lycopene biosynthesis pathway, GAL4-tsINT demonstrated superior performance, with its lycopene titer being 15.5-fold higher than that of GAL4M9 when using glycerol and galactose as carbon sources (YPGlyG) (Supplementary Fig. S7). In the sanguinarine pathway, the titer of GAL4-tsINT was 2.24 times higher than that of GAL4M9 (SEGlyG) (Fig. 1d of the revised manuscript). Overall, based on the performance across these three scenarios, we tentatively conclude that there remains some activity gap between GAL4M9 and GAL4-tsINT. While this discrepancy may not be as pronounced when initiating the expression of a single gene (*mCherry*), it becomes increasingly apparent when initiating the expression of multiple genes

(lycopene pathway). For the more complex sanguinarine pathway, considering additional bottlenecks and complex influences, the performance is not as robust as observed in the lycopene pathway. In conclusion, we have summarized the pros and cons of GAL4M9 and SIMTeGES-GAL4 in Supplementary Table S1.

Supplementary Table S1 The pros and cons of GAL4M9 and SIMTeGES-GAL4

Systems	GALM9	SIMTeGES-GAL4
Mechanisms	Temperature-sensitive mutant obtained by directed evolution	Inserting temperature-sensitive intein into the host proteins
Advantages	Lower leakage expression level	Stronger transcriptional activation activity Suitability for temperature-controlled expression of various target proteins in different host systems
Disadvantages	Temperature dependence is associated with a trade-off in transcriptional activation capacity, resulting in lower transcriptional activation activity Limited to regulating the expression of target genes under the control of GAL promoters in yeast	Slightly higher leakage expression level

In addition, the western-blot results in Figures 3e and 3f validate the direct correlation between protein activity and intein splicing with SIMTeGES for direct control of *mCherry* expression (SIMTeGES-*mCherry*). To avoid misinterpretation, we have included schematic diagrams for SIMTeGES-GAL80 and SIMTeGES-*mCherry* in the revised supplementary information (Supplementary Fig. S10). As pointed out by the reviewer, indeed, there is a band at 35 kDa (spliced form) for SIMTeGES-*mCherry* under non-permissive conditions. To more accurately characterize the activity and leakage level of *mCherry*-tsINT, we have supplemented the corresponding flow cytometry results (Supplementary Fig. S11). The leakage expression level of *mCherry*-tsINT under non-permissive conditions is only 5.21%, while the positivity rate under permissive conditions is 99.8%, further demonstrating that SIMTeGES exhibits strong activity and low leakage expression levels in different host proteins. We have incorporated these results into Lines 245-248 of the revised manuscript.

As for the observed discrepancy between western-blot (Fig. 2e) and flow cytometry (Supplementary Fig. S11) results, we speculate the main reason as the occurrence of intein splicing during sample preparation for western-blot analysis. As we know, intein splicing is a cofactor or energy-independent intramolecular process mainly through bond rearrangement. Although we expedited the sample preparation process for western-blot analysis as quickly as possible, it still takes longer than fluorescence detection (flow cytometry and fluorescence microplate reader), and protein samples under non-permissive conditions are likely to undergo partial splicing during this process (room temperature is 25 °C). Therefore, western-blot results can only serve as semi-quantitative references and are more suitable for proving the conclusion that temperature-dependent activity relies on intein splicing.

Supplementary Fig. S10 Schematic diagrams for SIMTeGES-GAL80 and SIMTeGES-mCherry.

Supplementary Fig. S11 Flow cytometry to further investigate the activity and leaky expression of mCherry-tsINT.

4. In Figure 6a the authors show a very nice example of metabolic engineering (or bottleneck identification) based on metabolite accumulation. However, the presentation is very basic. The authors should provide quantitative data (e.g. fold-changes) for the metabolites together with abundances of the respective enzymes. This would help to understand the systematic approach for changes in the pathway.

Thanks for the reviewer's comments. We agree with the reviewer that a more detailed description of the global pathway optimization in Fig. 6a is needed. Previously, this section was condensed due to content constraints. Here, we have included quantitative data for the intermediates, such as fold-changes, as outlined in Lines 412-413, 416-418, and 430-432 of the revised manuscript. Additionally, we conducted qPCR quantitative analysis on strains SAN223-4 and SAN231 to evaluate the abundance of each enzyme involved in the heterologous expression or overexpression of genes depicted in Fig. 6a (Supplementary Fig. S25). Our analysis revealed a significant increase in expression levels for the newly introduced heterologous genes *LmXFPK*, *EcoAROL*, and *BUP1*, as well as the endogenous genes *MET17*, *STL1*, *SOD1*, and *ADO1*, suggesting a corresponding increase in protein expression levels. Additionally, both *EcSTS* and *Ps6OMT*, genes in the sanguinarine pathway, showed a moderate increase in protein expression levels with the addition of an extra copy. These findings further validate the efficacy of our engineered modifications and have been incorporated into Lines 443-446 of the revised manuscript. We believe the additional data will provide a deeper understanding of the metabolic engineering optimization process and enhance the clarity of Fig. 6a.

Supplementary Fig. S25 Quantitative analysis of gene expression levels between strains SAN223-4 and SAN231 using qPCR. Data are presented as mean \pm s.e.m. (n = 3 biologically independent samples).

5. The authors use complex medium for their bioprocess for sanguinarine biosynthesis. At the time point of sanguinarine synthesis the main carbon source glycerol and galactose are depleted. Therefore, the authors should show which carbon source goes into sanguinarine biosynthesis. It is not clear if tyrosine fed to the cultures in Figure 6 or if tyrosine was derived from the complex medium.

Thanks for the reviewer's comments. We agree with the reviewer that providing additional clarity regarding the flux of carbon sources into sanguinarine biosynthesis is necessary. In our study, we primarily utilized 20 g L⁻¹ of galactose as the main carbon source in synthetic medium (SE), which undergoes assimilation to produce glucose-6-phosphate (G6P). Subsequently, phosphoenolpyruvate (PEP) and erythrose-4-phosphate (E4P) are generated via the pentose phosphate pathway (PPP) and glycolytic pathway, respectively, contributing to sanguinarine biosynthesis through the shikimate pathway, as illustrated in Supplementary Fig. S17b. Additionally, glycerol served as a secondary carbon source, converted to dihydroxyacetone phosphate (DHAP), thereby bolstering the supply of PEP. Given the comparatively lower assimilation rates in *S. cerevisiae*, we used 10 g L⁻¹ glycerol as non-inhibitory and non-inducing carbon source for seed culture. As shown in Fig. 6b, sanguinarine biosynthesis was initiated until all glycerol was consumed during fed-batch fermentation. In other words, glycerol was mainly used for cell growth, while galactose was used for both cell growth and sanguinarine biosynthesis. Therefore, we tentatively conclude that galactose serves as the main carbon source for sanguinarine biosynthesis.

Supplementary Fig. S17b Schematic representation of DEGs within central carbon metabolic pathways when glycerol and galactose are carbon sources, with upregulated genes in red and downregulated genes in green.

Regarding the effect of tyrosine, it's pertinent to clarify that our shake flask fermentation for sanguinarine biosynthesis utilized a synthetic medium SE, distinct from the complex medium YP. SE comprises 1.7 g L^{-1} yeast nitrogen base without ammonium and amino acids, 1.1 g L^{-1} monosodium glutamate, and Complete Supplement Mixture (CSM; 1.2 g L^{-1} ; MP Biomedicals), containing 100 mg L^{-1} of tyrosine, without additional supplementation. In the work titled "A yeast platform for high-level synthesis of tetrahydroisoquinoline alkaloids," a $2\times\text{SC}$ medium was used, with the Drop-out Medium Supplements (Millipore-Sigma) minus appropriate amino acids component including 152 mg L^{-1} of tyrosine. Consequently, only a minor portion of the tyrosine precursor for the sanguinarine pathway originates from the medium we employed, primarily sourced from galactose assimilation. Moreover, we conducted supplementary experiments to assess whether additional tyrosine supplementation at concentrations ranging from 50 to 300 mg L^{-1} could further enhance sanguinarine biosynthesis (Fig. R3). However, no discernible increase in production was observed. In other words, despite our medium containing a nominal amount of tyrosine within the typical range observed in other studies, it did not positively impact sanguinarine production. We speculate that the precursor availability upstream of the pathway is not the limiting step, indicating potential downstream bottlenecks that warranting further investigation.

Fig. R3: The effect of tyrosine concentration on sanguinarine titer in the culture medium. The abbreviations 50-Tyr, 100-Tyr, 150-Tyr, 200-Tyr, 250-Tyr, and 300-Tyr represent the additions of 50, 100, 150, 200, 250, and 300 mg L^{-1} of L -tyrosine to the culture medium, respectively.

Reviewers' Comments:

Reviewer #1:

Remarks to the Author:

The authors have addressed most of my concerns. Although the fluorinated sanguinarine could not be characterized by NMR due to the low yield of this product, the authors have supplemented substantial evidence using HRMS. Thus, its identity can be confirmed. It now can be accepted.

Reviewer #2:

Remarks to the Author:

The authors have addressed most of the reviewer's previous concerns, so the revised paper is much improved and is now in a good state.
I have no further comments.

Reviewer #3:

Remarks to the Author:

The authors have addressed all of my points in detail, with new experiments and well structured revisions in the text. This is now a very strong manuscript, I have no further points.